# $k$-NN as a Simple and Effective Estimator of Transferability

**Moein Sorkhei**  *sorkhei@kth.se*
*KTH Royal Institute of Technology, Stockholm, Sweden*
*Science for Life Laboratory, Stockholm, Sweden*

**Christos Matsoukas**  *matsou@kth.se*
*KTH Royal Institute of Technology, Stockholm, Sweden*
*Science for Life Laboratory, Stockholm, Sweden*

**Johan Fredin Haslum**  *jhaslum@kth.se*
*KTH Royal Institute of Technology, Stockholm, Sweden*
*Science for Life Laboratory, Stockholm, Sweden*

**Emir Konuk**  *ekonuk@kth.se*
*KTH Royal Institute of Technology, Stockholm, Sweden*
*Science for Life Laboratory, Stockholm, Sweden*

**Kevin Smith**  *ksmith@kth.se*
*KTH Royal Institute of Technology, Stockholm, Sweden*
*Science for Life Laboratory, Stockholm, Sweden*

**Reviewed on OpenReview:** *https://openreview.net/forum?id=hGlkjP1zHc*

## Abstract

How well can one expect transfer learning to work in a new setting where the domain is shifted, the task is different, and the architecture changes? Many transfer learning metrics have been proposed to answer this question. But how accurate are their predictions in a realistic new setting? We conducted an extensive evaluation involving over 42,000 experiments comparing 23 transferability metrics across 16 different datasets to assess their ability to predict transfer performance for image classification tasks. Our findings reveal that none of the existing metrics perform well across the board. However, we find that a simple $k$-nearest neighbor evaluation – as is commonly used to evaluate feature quality for self-supervision – not only surpasses existing metrics, but also offers better computational efficiency and ease of implementation.

## 1 Introduction

Transfer learning is a widely used technique for reusing knowledge learned in one domain – the *source* – to improve performance in another – the *target.* The question of whether transfer learning will be beneficial in a particular setting, and to what extent, is not always clear. Many metrics designed for this purpose – so-called transferability metrics – have been proposed. They typically focus on *only one* of three factors that influence the transfer learning process: *(1)* the domain distance – *i.e.* what happens when the target data differs from the source, *(2)* variations in task and task complexity – *i.e.* what happens when the target task differs from the source task, and *(3)* the choice of architectural design – *i.e.* what happens when the network architecture changes. We argue that a good transferability metric should simultaneously account for all these factors not only independently, but also in combination.

We further posit that, when predicting transferability, one should not only consider the accuracy in predicting the final performance of the transferred model in the new setting, but also the *ability to predict performance improvements* gained through the transfer learning process. The reasoning behind the former should be

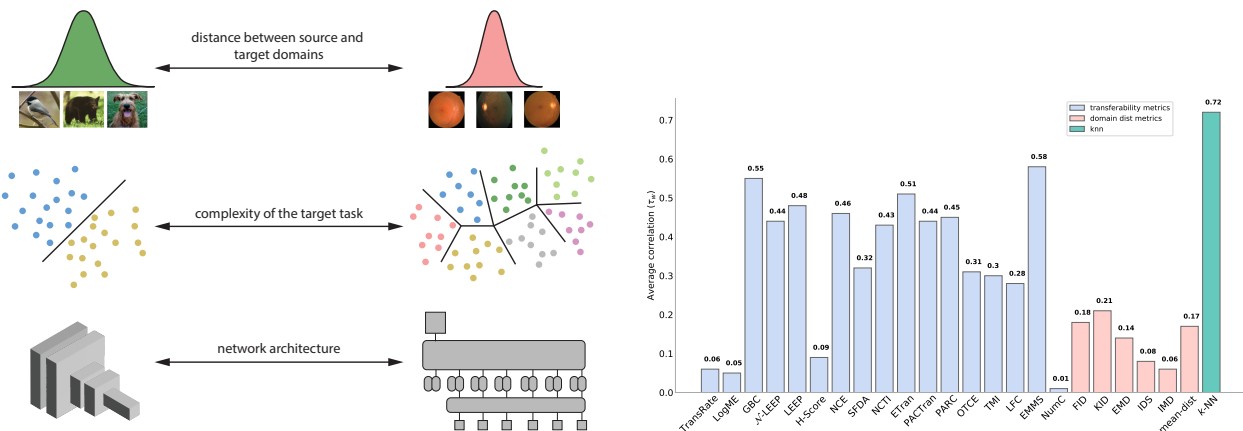

Figure 1: We investigate how well existing transferability metrics perform across three essential factors of transferability (Left): *1)* shifts between the source and target domains, *2)* changes in target task, and *3)* changes in network architecture. (Right) How effectively different metrics predict transferability, averaged across all the above settings. Surprisingly, we find that a simple nearest neighbor evaluation ($k$-NN) outperforms all existing metrics from the literature.

obvious – we are interested in knowing what the final performance of the model will be once we apply transfer learning. But we should also be interested in the latter – knowing how much benefit, if any, we will see from transferring knowledge from the source domain (*i.e.* the pretrained weights) as compared to learning from a random initialization. Carelessly transferring from an inappropriate data source can lead to biased or even inferior performance (Torralba & Efros, 2011; Matsoukas et al., 2022; Zhang et al., 2022; Raghu et al., 2019; He et al., 2019). If random initialization can do a similar or better job, shouldn't we want to know?

In this work, we conducted an extensive evaluation involving over 42,000 experiments comparing 23 transferability metrics across 16 different datasets to assess their ability to predict transfer performance for image classification. We establish desiderata (the key features or qualities) of a good transferability metric – it should accurately predict performance in the target setting after transfer learning with robustness to domain shifts, changes in the task, and architecture changes (see Figure 1). Furthermore, the metric should be able to predict both the final performance of the transferred model as well as the gains anticipated from transfer learning. Our experiments show that *none of the existing metrics consistently performs well across these criteria.*

In pursuit of a robust metric, we explore the application of nearest neighbor evaluation ($k$-NN) – a commonly used technique in self-supervision to evaluate the quality of learned feature representations – within the context of transferability estimation. We discover that *simple nearest neighbor evaluation surpasses all the metrics from the literature* (see Figure 1). Furthermore, $k$-NN possesses attractive properties such as being cheap, optimization-free, and closely resembling the target classification task. Additionally, its straightforward implementation adds to its appeal. Together, these factors make $k$-NN a compelling choice as a transferability metric.

## 2 Methods

In this work, we ask: *to what extent are established metrics of transferability useful across changes in domain, task complexity, and architecture? Is there a single metric that performs well across all scenarios?* We begin by describing the metrics used in our study.

**Transferability metrics** Transferability metrics are designed to predict the efficacy of transfer learning, taking into account task and architectural differences. Below, we briefly describe 17 such metrics, starting with metrics that model the relationship between source and target tasks:

- **NCE** (Tran et al., 2019) focuses on task differences and estimates transferability by calculating the negative conditional entropy between the source and target labels.

- **LEEP** (Nguyen et al., 2020) models the joint empirical distribution between the source labels, as predicted by the pre-trained model, and the target labels and calculates the average log-likelihood of the target labels given the predicted source labels.

- **OTCE** (Tan et al., 2021) models task and domain differences. Domain difference is measured using optimal transport between source and target domains, and task difference is measured using conditional entropy between source and target labels.

- **NumC** (Agostinelli et al., 2022) we additionally utilize the inverse of the number of classes as a trivial transferability metric, implying that a lower number of classes signifies higher transfer performance.

In addition, we consider transferability metrics designed to predict transfer efficacy under architectural changes:

- **TransRate** (Huang et al., 2022) estimates the mutual information between the target embeddings, extracted from the pretrained model, and the target labels.

- **LogME** (You et al., 2021) models transferability through estimating the maximum value of the target label evidence given the target features extracted from the pre-trained model.

- **GBC** (Pándy et al., 2022) models each target class with a Gaussian distribution in the pre-trained feature space and calculates the Bhattacharyya distance Bhattacharyya (1946) between all pairs of classes.

- $\mathcal{N}$-**LEEP** (Li et al., 2021) is an extension of LEEP with the difference that it removes the classification head of the pretrained model and instead fits a Gaussian Mixture Model (GMM) on the target embeddings and uses such cluster assignments in place of source labels to compute the LEEP score.

- **H-score** (Bao et al., 2019) estimates the feature redundancy and the inter-class variance of the target embeddings extracted from the pre-trained model. Low feature redundancy and high inter-class variance (high H-score) suggests good transfer learning performance.

- **SFDA** (Shao et al., 2022) projects target features, extracted from the pre-trained model, into a space with high class separability and employs Bayes' rule to assign instances to different classes, assuming a normal distribution for each class.

- **NCTI** (Wang et al., 2023) gauges the proximity between the current state of the pre-trained model and its hypothetical state in the terminal stage of fine-tuning. Closer proximity signifies superior transfer performance.

- **ETran** (Gholami et al., 2023), akin to SFDA, projects target features into a class-separable space and employs Bayes' rule, following the assumption of normal distribution for each class. It also computes an energy score on target features to determine whether the target data is in- or out-of-distribution for the pre-trained model.

- **PACTran** (Ding et al., 2022) assesses transferability by quantifying the pre-trained model's generalization to the target task, utilizing the PAC-Bayes bound (McAllester, 1998; Germain et al., 2009).

- **PARC** (Bolya et al., 2021) computes the dissimilarity between pairwise target features and the dissimilarity between corresponding pairwise labels. It then calculates the correlation between these two measures, operating on the premise that images with distinct labels should exhibit dissimilar feature representations.

- **TMI** (Xu & Kang, 2023) assesses transferability through measuring intra-class variance of the target features, arguing that higher intra-class variance leads to higher transfer performance.

- **LFC** (Deshpande et al., 2021) uses a linearized framework to approximate finetuning and measures Label-Feature correlation for estimating transferability.

- **EMMS** (Meng et al., 2023) estimates transferability by establishing a linear regression between target features and labels derived from a foundation model.

**Domain distance metrics**  The distance between two domains is often crucial for successful transfer learning – a larger domain distance is associated with a reduced net gain from transfer learning.

While not explicitly designed to predict transferability, we consider 5 widely recognized methods for measuring domain distance:

- **Fréchet Inception Distance** (Heusel et al., 2017; Fréchet, 1957): The FID score models each dataset with a Gaussian distribution and measures the distance between the two datasets using the Fréchet distance.

- **Earth Mover's Distance** (Cui et al., 2018; Rubner et al., 2000): EMD views each dataset as a set of clusters represented by the class mean features, weighted by the number of images per class, and calculates the Wasserstein distance between the two sets of clusters.

- **Kernel Inception Distance** (Bińkowski et al., 2018; Gretton et al., 2012): KID is a non-parametric distance measure based on the maximum mean discrepancy of the features after applying a polynomial kernel.

- **Image Domain Similarity** (Mensink et al., 2021): IDS is an asymmetric distance measure that takes samples from the source and target domains and calculates the average of the distances between each target sample to its closest source sample.

- **Intrinsic Multi-scale Distance** (Tsitsulin et al., 2019): The IMD metric measures the discrepancy between two distributions by approximating the underlying data manifolds and the distance is calculated using the spectral Gromov-Wasserstein inter-manifold distance.

- **Mean-dist**: In addition to the 5 metrics described above, we also compute the the average of all domain distance metrics, denoted as "mean-dist".

**$k$-NN as a transferability metric**  In this work, we propose to use nearest neighbor evaluation ($k$-NN) as a metric of transferability. Drawing inspiration from its use in self-supervised tasks to measure the pretrained model's ability to partition the data based on its relevance to the target task (Caron et al., 2021; Wu et al., 2018; He et al., 2020), we employ $k$-NN to predict transferability.

Specifically, we split the training set (S) of the target in two disjoint subsets S1, S2 of 80%-20% of the training set. Subsequently, $k$-NN classification was performed on S2 using $k$ nearest neighbors from S1. The resulting $k$-NN accuracy served as the transferability score (to ensure reliability, we repeated the same procedure with 3-fold cross-validation on the training set, yielding identical results). Following common practice, cosine similarity of extracted image features was used to measure distance between data points. We chose a default value of $k = 200$, and found that its performance remained consistent w.r.t. $k$ (Figure 9b in the Appendix).

## 3 Experiments

Our experiments are conducted as follows. Given a source, target, and architecture, we obtain scores from 23 metrics applied to the training set of the target data. We then use transfer learning to train networks on the target dataset and measure the absolute and relative performance. Transfer learning involves initially training a network on a source domain, followed by fully fine-tuning it on the target domain. To assess the performance of a metric, we compute the correlation between its transferability score and the observed performance after transfer learning. In addition to the main experiments, we measure transferability prediction performance isolated for domain shift, change of task, and change of architecture.

**Datasets**  We apply transfer learning across a diverse set of 16 image classification datasets. For the source domains, we selected IMAGENET (Deng et al., 2009), iNat2017 (Van Horn et al., 2018), PLACES365 (Zhou et al., 2017), and NABirds (Van Horn et al., 2015). As target datasets, we include well-known benchmarks such as CIFAR-10 and CIFAR-100 Krizhevsky et al. (2009), Caltech-101 (Fei-Fei et al., 2004), Caltech-256 (Griffin et al., 2007), StanfordDogs (Khosla et al., 2011), Aircraft (Maji et al., 2013), NABirds Van Horn et al. (2015), Oxford-III Pet (Parkhi et al., 2012)), SUN397 (Xiao et al., 2010), DTD (Cimpoi et al., 2014), AID (Xia et al., 2017), and APTOS2019 Karthik (2019). These datasets cover super-ordinate object recognition, fine-grained classification, scene recognition, texture classification, aerial imagery, and medical imaging (details in Table 3 in the Appendix).

**Architectures**  To investigate the effect of architectural changes, we utilize a set of 11 architectural variations drawn from five model families. In detail, we utilize ResNet-50 (He et al., 2016), Inception-V3 (Szegedy et al., 2016), DenseNet-121 (Huang et al., 2017), MobileNet-V2 (Sandler et al., 2018), RegNetY-3.2GF (Radosavovic et al., 2020), ResNeXt-50 (Xie et al., 2017), WideResNet-50-2 (Zagoruyko & Komodakis, 2016), DeiT-Small (Touvron et al., 2021), Swin-Tiny (Liu et al., 2021), PiT-Small (Heo et al., 2021), and PVT-Small (Wang et al., 2021). Domain distance metrics rely on a network to extract features upon which the distance is calculated. For this, we consider five representative IMAGENET-pretrained architectures: ResNet-50, Inception-V3, DenseNet-121, DeiT-Small, and Swin-Tiny.

**Implementation details**  For each dataset, either the official train/val/test splits were used, or we made the splits following (Kornblith et al., 2019). Images were normalized and resized to $256 \times 256$, after which augmentations were applied: random color jittering, random horizontal flip and random cropping to $224 \times 224$ of the rescaled image. The Adam optimizer (Kingma & Ba, 2014) was used for CNNs and AdamW (Loshchilov & Hutter, 2017) for ViT-based architectures, and the training of models was done using PyTorch (Paszke et al., 2019). We provide details regarding the training procedure in the Appendix.

**What performance should we measure?**  When applying transfer learning, one may wish to predict:

- **absolute transfer performance** – the performance of the transferred model measured in the target domain, in absolute terms;

- **relative transfer performance** – how much performance improvement was gained in the target domain as a result of transfer learning.

The absolute transfer performance is essential to measure how well the transferred model can address the target task/setting. Throughout this work, absolute transfer performance is denoted as $perf(P)$.

While $perf(P)$ gives a prediction about the expected performance, it offers limited insight into the effectiveness of the knowledge transfer. Consider, for example, a task that is easy to solve. Even though the $perf(P)$ may be high in absolute terms, it may not necessarily imply that transfer learning was beneficial – in fact, it often isn't (He et al., 2019; Neyshabur et al., 2020; Raghu et al., 2019). We must also consider the performance gained as compared to the same model initialized with random weights (He et al., 2015), denoted as $perf(RI)$. Following Neyshabur et al. (2020), we define the net gain from transfer learning as the *Relative Transfer Performance* (RTP) between a pretrained network $P$ and a randomly initialized model $RI$, expressed as: $\text{RTP} := \frac{perf(P) - perf(RI)}{perf(P)}$. When applying this to predictions from transferability metrics, we adopt a similar formulation $\text{RTP}_{\text{P}} = \frac{est(P) - est(RI)}{est(P)}$, where $est(P)$ and $est(RI)$ denote the transferability scores for the pretrained and randomly-initialized models respectively. We also introduce *transfer gap*[1]. as the complement of RTP, represented by $\text{TG} := 1 - \text{RTP}$.

We judge the performance of the transferability metrics by checking how well the transferability scores[2] correlate with the performance observed from actually applying transfer learning (both absolute or relative).

---

[1]The complement, TG, captures the inverse relationship between domain distance and net benefits from transfer learning. It is used when measuring the correlation between domain distance metrics and the gain from transfer learning

[2]We invert the domain distance estimates to predict absolute transfer performance, accounting for their inverse relationship.

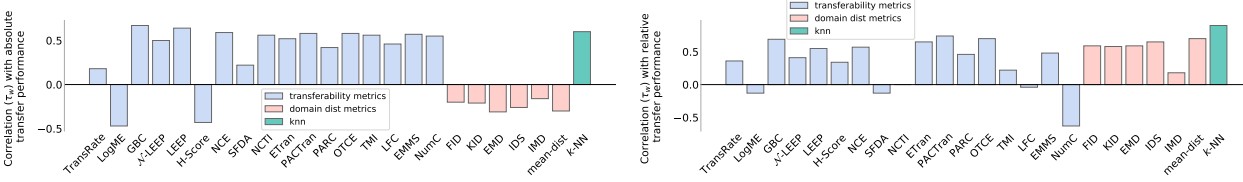

(a) Correlations with absolute transfer performance.  (b) Correlations with relative transfer performance.

Figure 2: Effectiveness of different metrics *when both target domain and task complexity change*. Results are averaged across four source datasets described in the text. Full break-down of results appears in Tables 4 and 5 in the Appendix.

This is quantified using weighted Kendall's tau ($\tau_w$) (Vigna, 2015), in line with previous studies (You et al., 2021; Shao et al., 2022; Wang et al., 2023; Pándy et al., 2022). Weighted Kendall's tau measures the correlation between ranked lists by assessing the number of correctly ordered pairs, particularly emphasizing the accurate ranking of high-performing models. We further employ other common evaluation measures in Figure 10 of the Appendix. We use accuracy to compute transfer performance for all tasks unless otherwise specified.

## 4 Results

We begin by reporting our main results, where the factors of transferability are considered simultaneously, followed by experiments isolating each factor.

**Which metrics are general predictors of transferability?** Our main results consider the scenario where both the target domain and the target task change from the source. In this setup, a network is pre-trained on each of the four source datasets and subsequently transferred to each of the 12 target datasets (as detailed in previous section). For each network, we compute both the absolute and relative transfer performance, and then measure the correlation between these and the transferability metrics (including domain distance metrics).

The results are reported in Figure 2. Metrics designed to measure domain distance appear in pink, while metrics designed to estimate transferability appear in blue. When predicting absolute transfer performance, the transferability metrics generally outperform the domain distance metrics, with the exception of LogME and H-score (Figure 2a). Among the transferability metrics, GBC stands out as the top-performing metric. Domain distance metrics appear to have a negative correlation with absolute transfer performance. This may be attributed to their inability to consider task complexity or network architecture, two crucial factors directly influencing absolute transfer performance.

When examining Relative Transfer Performance (RTP), as depicted in Figure 2b, the picture changes. In contexts where the focus is on quantifying the benefits gained from transfer learning, domain distance metrics exhibit good performance (with the exception of IMD). Mean-dist, while being less effective than PACTran, proves to be particularly adept at assessing the net gains from transfer learning, albeit at a high computational cost.

Setting aside *k*-NN for now, several metrics fail to perform well in either or both absolute and relative prediction. GBC is the leading metric for predicting absolute transfer performance, while PACTran excels in terms of predicting relative transfer performance. This suggests that there is no universally superior metric in this setting – the most suitable choice may vary based on the intended use and the source dataset, as elaborated in Tables 4, 5 in the Appendix.

**Which metric is better when only the domain changes?** We next consider how the metrics perform when only the domain distance changes – the target task and network architecture remain the same. In this experiment, we use IMAGENET as the source domain and we fine-tune models on DomainNet (Peng et al., 2019). DomainNet comprises six distinct domains, each featuring the same set of 345 categories. As an additional experiment, we set each individual domain, within DomainNet, as the source and use as targets

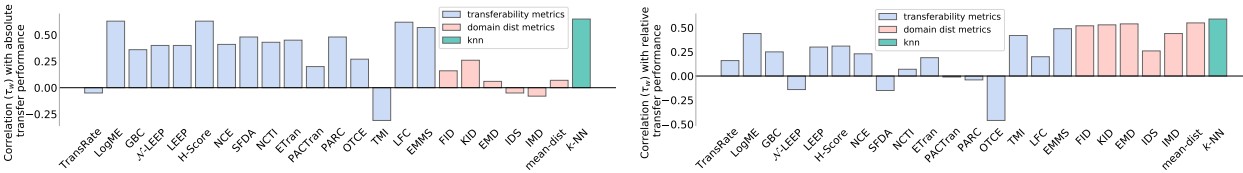

(a) Correlations with absolute transfer performance.

(b) Correlations with relative transfer performance.

Figure 3: Effectiveness of different metrics *when only the target domain changes*. Results are averaged over the seven configurations described in the text. Full break-down of results appears in Tables 6 and 7 in the Appendix.

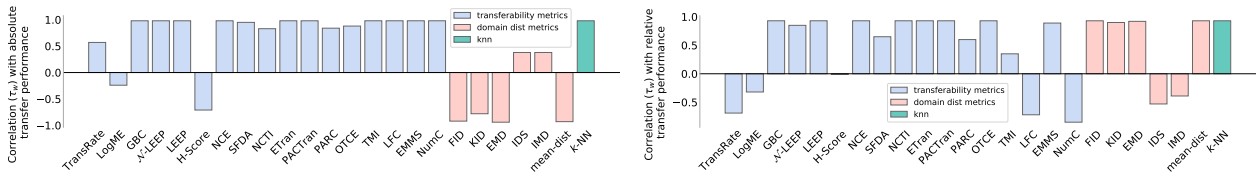

(a) Correlations with absolute transfer performance.

(b) Correlations with relative transfer performance.

Figure 4: Effectiveness of transferability metrics *when only target task complexity changes*. Results are averaged over NABirds, SUN397, and Caltech-256. Full results appear in Tables 8 and 9 in the Appendix.

the remaining domains. Condensed results appear in Figure 3. Detailed results are in Tables 6, 7 in the Appendix.

In terms of absolute transfer performance (Figure 3a), LogME and H-score, which exhibited poor correlation in the previous setting, now emerge as the top-performing metrics. Interestingly, GBC is no longer the leading metric for predicting absolute transfer performance – in fact, it is among the weakest. The correlation between the domain distance metrics (in pink) is better than in Figure 2a, possibly because domain distance is the only varied factor. Perhaps surprisingly, transferability metrics perform better at predicting *absolute* transfer performance in this scenario (with the exception of TransRate and TMI).

When we consider relative transfer performance in Figure 3b, the domain distance metrics show their superiority over the transferability metrics. EMD emerges as the top performer, albeit only by a narrow margin compared to FID and KID. PACTran, which was the best metric to predict relative transfer performance in the previous setting, exhibits negative correlation in this setting. $k$-NN's superior performance will be discussed later.

**Which metric is better when only the task changes?** Next, we consider the case of changing the task. Using IMAGENET as the source, we apply transfer learning to subsets of NABirds, SUN397, and Caltech-256. More specifically, we vary the task complexity by varying the number of classes (as a proxy) in accordance with Deng et al. (2010); Konuk & Smith (2019). In detail, we sequentially remove classes from the target dataset to create target tasks of varying complexity. The results are provided in Figure 4 as averaged correlations, and the detailed results appear in Tables 8, 9 in the Appendix. Again, we refrain from discussing $k$-NN until a later section.

In terms of absolute transfer performance, it is evident from Figure 4a that the transferability metrics (blue) are generally superior to the domain distance metrics (pink). Domain distance metrics exhibit a strong negative correlation with absolute performance (except from IDS and IMD). Interestingly, LogME and H-score, the best metrics to predict absolute transfer performance in the previous setting (Figure 3a), now appear ineffective.

The results on relative transfer performance (Figure 4b) follow a similar pattern as in Figure 4a for the transferability metrics (except for TransRate and LFC). However, the domain distance measures reveal a contrasting pattern. In general, both transferability and domain distance metrics are either very highly

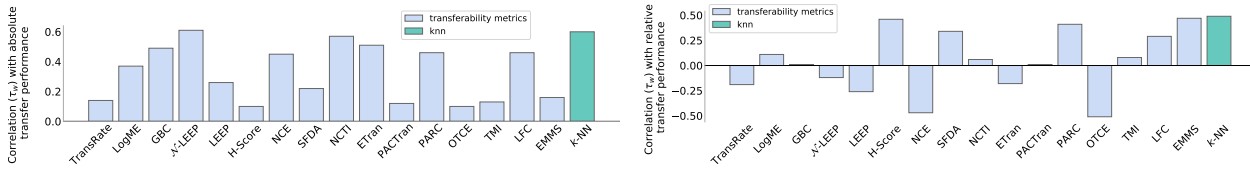

(a) Correlations with absolute transfer performance.  (b) Correlations with relative transfer performance.

Figure 5: Effectiveness of transferability metrics *when only network architecture changes*. Results for each metric denote average correlation over 12 downstream datasets. Full results for individual datasets appear in Tables 1 and 2.

correlated with relative transfer performance, or negatively correlated. While there is no clear winner in this test, there are some clear losers (TransRate, LogME, H-score, LFC, IDS, and IMD).

**Which metric is better when only the network architecture changes?** We also examine the effect of isolating network architecture as a factor of transferability. We consider 11 distinct network architectures starting from IMAGENET pre-training and perform transfer learning on each of the 12 targets. We compute correlations between the transferability metrics and the absolute and relative transfer performance. In this setting, we omit the domain distance metrics since they are not applicable in this setup (as both the target domain and task remain the same). The results are presented in Figure 5 and we provide detailed, per-dataset correlations in Tables 1 and 2.

When it comes to predicting absolute transfer performance in Figure 5a, $\mathcal{N}$-LEEP outperforms the other transferability metrics from the literature, followed by NCTI. This result is unsurprising, as both of these metrics are specifically designed to characterize architectural variations. For relative transfer performance, shown in Figure 5b, nearly all the existing transferability metrics fail, with many exhibiting a negative correlation. EMMS, followed by H-score and PARC, stand out by significantly outperforming the other metrics, suggesting that these are the only reliable transferability metrics when predicting relative transfer performance when architecture is the only factor that changes.

**$k$-NN as a reliable predictor of transferability** Finally, we comment on the transferability predictions of $k$-NN. In Figures 2 through 5, $k$-NN results were reported in turquoise. In Figure 2, which showed the general ability to predict transferability, $k$-NN performed remarkably well – it performs comparably to the best transferability metrics when measuring absolute performance, and outperforms all other metrics in terms of relative transfer performance by a wide margin. This trend persists when the individual factors are isolated (Figures 3 - 5). Notably, when isolating domain change (Figure 3), $k$-NN outperforms all the other metrics in predicting both absolute and relative transfer performance.

Compared to the methods from the literature, $k$-NN performed consistently well. In fact, $k$-NN was the only method among the 24 to maintain a positive correlation throughout all the experiments. Combined with its optimization- and hyperparameter-free nature, as well as its low computational cost and ease of implementation, our findings suggest that $k$-NN can be seamlessly employed as a reliable transferability metric.

**Computational efficiency** The total running time versus the overall performance of the metrics is depicted in Figure 9a in the Appendix. It is evident that $k$-NN provides computational cost comparable to the fastest metrics while, at the same time, outperforming them.

## 5 Discussion

We argue that a metric for transferability should accurately predict both absolute and relative transfer performance, as each answer different but important questions (*i.e.* what performance level can I expect? vs. what benefit will transfer learning provide?) Not only that, but an ideal metric should also be reliable across different transfer learning scenarios: changes in the domain, task, or architecture.

Table 1: Effectiveness of transferability metrics to predict *absolute transfer performance* when *only network architecture changes*, shown for individual target datasets.

|  | Dogs | SUN397 | DTD | CIF-100 | CIF-10 | Cal-256 | Cal-101 | Pet | Aircraft | Birds | APTOS | AID |
|---|---|---|---|---|---|---|---|---|---|---|---|---|
| TransRate | 0.353 | 0.183 | 0.261 | 0.040 | 0.234 | -0.066 | -0.109 | 0.202 | 0.101 | 0.378 | 0.136 | 0.032 |
| LogME | 0.578 | 0.090 | 0.023 | 0.535 | 0.684 | 0.324 | 0.751 | 0.576 | 0.263 | 0.257 | 0.589 | -0.260 |
| GBC | 0.765 | 0.702 | 0.299 | 0.830 | 0.699 | 0.293 | 0.219 | 0.664 | -0.048 | 0.676 | 0.478 | 0.311 |
| $\mathcal{N}$-LEEP | 0.783 | 0.835 | 0.647 | 0.821 | 0.574 | 0.816 | 0.537 | 0.940 | 0.257 | 0.392 | 0.420 | 0.330 |
| LEEP | 0.575 | 0.648 | -0.046 | 0.758 | 0.341 | 0.494 | 0.192 | -0.152 | 0.462 | 0.187 | 0.080 | -0.362 |
| H-Score | 0.545 | 0.241 | -0.205 | 0.218 | 0.288 | 0.064 | 0.085 | -0.096 | 0.441 | -0.177 | -0.012 | -0.161 |
| NCE | 0.712 | 0.711 | 0.442 | 0.904 | 0.744 | 0.842 | 0.540 | 0.771 | -0.173 | 0.359 | 0.254 | -0.657 |
| SFDA | 0.530 | -0.305 | -0.368 | 0.419 | 0.623 | 0.119 | 0.077 | -0.352 | 0.556 | -0.160 | 0.800 | 0.640 |
| NCTI | 0.782 | 0.883 | 0.613 | 0.801 | 0.682 | 0.444 | 0.526 | 0.728 | 0.080 | 0.405 | 0.428 | 0.505 |
| ETran | 0.790 | 0.930 | 0.773 | 0.866 | 0.682 | 0.763 | 0.021 | 0.656 | 0.220 | -0.029 | -0.170 | 0.658 |
| PACTran | -0.079 | 0.334 | 0.017 | 0.155 | 0.024 | 0.062 | 0.032 | -0.305 | 0.703 | 0.088 | 0.212 | 0.223 |
| PARC | 0.803 | 0.631 | 0.173 | 0.813 | 0.673 | 0.125 | 0.410 | 0.556 | 0.008 | 0.716 | 0.309 | 0.357 |
| OTCE | 0.398 | 0.235 | 0.003 | 0.253 | -0.191 | 0.512 | 0.668 | 0.651 | -0.403 | -0.480 | -0.210 | -0.194 |
| TMI | 0.158 | 0.250 | 0.237 | 0.054 | 0.037 | 0.369 | 0.043 | 0.050 | 0.733 | -0.149 | -0.072 | -0.166 |
| LFC | 0.663 | 0.700 | 0.628 | 0.730 | 0.502 | 0.571 | 0.257 | 0.580 | -0.372 | 0.642 | -0.141 | 0.744 |
| EMMS | 0.566 | -0.101 | -0.351 | 0.450 | 0.684 | 0.010 | 0.393 | 0.236 | 0.594 | 0.079 | -0.261 | -0.437 |
| $k$-NN | 0.878 | 0.927 | 0.786 | 0.836 | 0.574 | 0.819 | 0.428 | 0.343 | 0.355 | 0.454 | 0.473 | 0.309 |

Table 2: Effectiveness of transferability metrics to predict *relative transfer performance* when *only network architecture changes*, shown for individual target datasets.

|  | Dogs | SUN397 | DTD | CIF-100 | CIF-10 | Cal-256 | Cal-101 | Pet | Aircraft | Birds | APTOS | AID |
|---|---|---|---|---|---|---|---|---|---|---|---|---|
| TransRate | -0.332 | 0.046 | -0.402 | 0.242 | 0.229 | -0.251 | -0.515 | -0.343 | -0.059 | -0.524 | -0.507 | 0.094 |
| LogME | 0.344 | 0.167 | 0.121 | 0.307 | -0.098 | 0.196 | -0.060 | 0.266 | 0.341 | -0.104 | -0.075 | -0.061 |
| GBC | 0.325 | -0.286 | 0.347 | -0.322 | -0.112 | -0.431 | 0.076 | 0.185 | 0.649 | -0.540 | 0.095 | 0.101 |
| $\mathcal{N}$-LEEP | 0.310 | 0.168 | -0.471 | -0.530 | -0.630 | 0.387 | -0.598 | -0.106 | -0.226 | 0.328 | 0.248 | -0.276 |
| LEEP | -0.022 | -0.048 | -0.492 | -0.257 | -0.225 | -0.083 | -0.189 | -0.512 | -0.647 | -0.336 | 0.038 | -0.319 |
| H-Score | 0.704 | 0.273 | 0.104 | 0.416 | 0.629 | 0.767 | 0.527 | -0.018 | 0.877 | 0.657 | 0.104 | 0.417 |
| NCE | -0.431 | -0.301 | -0.725 | -0.790 | -0.435 | -0.606 | -0.722 | -0.464 | -0.642 | -0.297 | 0.326 | -0.508 |
| SFDA | 0.383 | 0.215 | 0.547 | 0.165 | 0.149 | 0.391 | 0.290 | 0.447 | 0.460 | 0.385 | 0.273 | 0.388 |
| NCTI | -0.204 | -0.020 | 0.036 | 0.211 | -0.127 | 0.573 | 0.234 | 0.221 | -0.068 | -0.238 | -0.269 | 0.418 |
| ETran | -0.055 | -0.291 | -0.401 | -0.658 | -0.668 | -0.148 | 0.249 | 0.055 | -0.105 | -0.204 | 0.434 | -0.409 |
| PACTran | -0.025 | 0.141 | -0.110 | -0.214 | -0.176 | 0.112 | 0.297 | 0.317 | -0.033 | -0.312 | 0.338 | -0.271 |
| PARC | 0.800 | 0.534 | 0.372 | 0.526 | 0.259 | 0.762 | -0.252 | 0.673 | 0.840 | 0.524 | -0.154 | 0.073 |
| OTCE | -0.328 | -0.388 | -0.608 | -0.385 | -0.636 | -0.591 | -0.610 | -0.404 | -0.531 | -0.654 | -0.569 | -0.374 |
| TMI | -0.157 | -0.183 | 0.095 | -0.153 | 0.091 | -0.015 | 0.491 | 0.112 | 0.077 | 0.199 | 0.060 | 0.289 |
| LFC | 0.403 | 0.258 | 0.165 | 0.419 | -0.005 | 0.100 | 0.680 | 0.261 | 0.393 | 0.181 | 0.347 | 0.252 |
| EMMS | 0.520 | 0.600 | 0.610 | 0.430 | 0.370 | 0.570 | 0.410 | 0.470 | 0.440 | 0.410 | 0.440 | 0.350 |
| $k$-NN | 0.696 | 0.608 | 0.388 | 0.431 | 0.520 | 0.872 | 0.042 | 0.652 | 0.464 | 0.504 | 0.077 | 0.581 |

**Which metrics should we use when we want to predict absolute transfer performance?** If one cares about the final performance of the transferred model, irrespective of any gain/loss from transfer learning, our experiments show that transferability metrics are superior to domain distance metrics, but the choice of the best metric is not clear. While GBC is best when the domain and task change together (Figure 2a), LogME and H-score demonstrate superior performance when domain changes in isolation (Figure 3a). When only the task changes, NCE, LEEP, $\mathcal{N}$-LEEP, ETran, and PACTran emerge as the top-performing metrics (Figure 4a). When considering variations in network architecture, $\mathcal{N}$-LEEP stands out as the most effective metric (Figure 5a).

Domain distance metrics prove deficient, often anti-correlated with absolute transfer performance. This, combined with their inability to adequately account for task and architectural differences renders them unsuitable estimators of absolute transfer performance.

$k$-NN, on the other hand, consistently demonstrates good performance in every scenario – comparable to or better than the best transferability metrics. When the domain changes in conjunction with the task, $k$-NN is comparable to GBC. When domain changes in isolation, $k$-NN stands out as the top-performing metric. When task variation is considered, $k$-NN is on par with NCE, LEEP, $\mathcal{N}$-LEEP, ETran, and PACTran. Considering only architectural differences, it performs comparably against $\mathcal{N}$-LEEP while outperforming all the other metrics.

**Which metrics are better at predicting the relative transfer performance?** While the ability to predict the end performance of transferred models is undeniably valuable, it leaves unanswered a critical question: *how much did transfer learning actually help?* Our investigations reveal that existing transferability metrics do not provide a satisfactory answer to this question. Transferability metrics often failed to yield consistent positive correlations when domain distance, tasks, or architecture were isolated. Domain distance metrics, on the other hand, exhibited mostly a positive correlation with the relative transfer performance. When only the domain changes, they were superior to transferability metrics (Figure 3b), on par with them when only the task changes (Figure 4b), and slightly better when task changes along with the domain (Figure 2b).

However, not all domain distance metrics are equally adept at estimating these gains. IMD and IDS appear to falter as predictors when individual factors vary (Figures 3b, 4b), while IDS seems reliable when all factors are simultaneously altered (Figure 2b). The average of all domain distances (mean-dist) appears more reliable than any of the individual metrics (Figures 2b, 3b, 4b), albeit with a significant computational overhead.

$k$-NN consistently outperforms both transferability and domain distance metrics when measuring relative transfer performance across all settings (Figures 2b, 3b, 4b). Considered together with its consistently good ability to predict absolute transfer performance (Figures 2a, 3a, 4a), $k$-NN is clearly the most reliable transferability metric.

**Why existing metrics fail to meet all criteria?** Our analysis concludes that, aside from $k$-NN, there is no existing metric that consistently satisfies the criteria for predicting transferability. While transferability metrics are generally capable of modeling absolute transfer performance, they tend to be unreliable when the experimental setting changes. This inconsistency is perhaps unsurprising, given that existing transferability metrics primarily concentrate on individual factors within transfer learning. Most are designed to consider variations in task and architecture. It is then perhaps not surprising that these metrics may fall short when confronted with changes in domain distance. On the other hand, domain distance metrics are explicitly crafted to gauge data similarity. This design inherently limits their ability to accommodate the task or the architecture.

**On the success of $k$-NN as a transferability metric** Throughout our experiments, $k$-NN demonstrated the ability to reliably predict absolute and relative transfer performance. It performed consistently well when considering the factors (domain, task, architecture) both in combination and in isolation.

But why is $k$-NN so effective? To begin with, $k$-NN classification (Fix, 1985) is an optimization-free approach that measures separability in the target domain using pretrained weights from the source domain. The intuition is that, if the source-pretrained weights already do a good job of separating the target classes, even before they are adapted using fine-tuning, there is a good chance the knowledge transfer will be beneficial. $k$-NN is a cheap proxy for performing the fine-tuning on the target task/domain – which makes it directly useful as a transferability metric. Furthermore, the simplicity of the method may add to its reliability, as other transferability metrics use approaches in a similar spirit but are sensitive to hyperparameters. In addition, $k$-NN shares desired properties of other prominent transferability metrics, such as GBC and $\mathcal{N}$-LEEP. However, $k$-NN eliminates the need for pre-defined heuristics, such as the assumption of a Gaussian distribution in the case of GBC. Furthermore, unlike $\mathcal{N}$-LEEP, it does not require any form of training on the target data to assess transferability.

**Related work** Estimating the potential benefits of transfer learning is not trivial. While we advocate that a transferability metric must be *simultaneously* robust to domain shifts, task changes, and different architectures, several prior works have identified these individual factors as important (Yosinski et al., 2014; Azizpour et al., 2015; Kornblith et al., 2019). There is a large body of work on measuring distance between domains. Many approaches involve fitting distributions to data samples and then comparing the distributions, such as Fréchet Inception Distance (Heusel et al., 2017; Fréchet, 1957), Kernel Inception Distance (Bińkowski et al., 2018; Gretton et al., 2012), and Intrinsic Multi-scale Distance (Tsitsulin et al., 2019). While these works focus solely on the problem of measuring domain distance and ignore the problem of predicting transferability, some others do consider transferability, including Earth Mover's Distance (Cui et al., 2018; Rubner et al., 2000), Image Domain Similarity (Mensink et al., 2021), and Optimal Transport Dataset Distance (Alvarez-Melis & Fusi, 2020). Other works have focused on the impact of the architectural and task differences when

employing transfer learning. Some of them such as DEPARA (Song et al., 2020), and its variant (Song et al., 2019), model task differences assuming there exits a trained model for each task. Other works such as Taskonomy (Zamir et al., 2018) and Task2Vec (Achille et al., 2019) require re-training in order to model task relations. Others, like NCE (Tran et al., 2019) and LEEP (Nguyen et al., 2020) model the relation between source and target labels, and only a few, such as OTCE (Tan et al., 2021) also consider the similarity of the domains but requires learning of auxiliary tasks, making it less practical. Finally, several prior works have explored predicting transferability of pretrained models, focusing on the effect of architectural differences within transfer learning. These include TransRate, LogME, GBC, $\mathcal{N}$-LEEP, H-score, SFDA, ETran, NCTI, PACTran , PARC, TMI , LFC, and EMMS.

**Contributions w.r.t previous works** Our work brings important contributions on top of relevant prior works. Unlike prior works, we explicitly consider and analyze the individual factors of transferability, offering deeper insights into metric robustness. We introduce RTP and emphasize its significance in additional to absolute transfer performance, revealing new insights. Our broader evaluation includes a comprehensive set of domain distance metrics, showing instances where these metrics surpass transferability metrics (Figure 3b). We consider significantly more metrics (24) compared to all existing works: 6 in Agostinelli et al. (2022) 9 in Bolya et al. (2021), and 7 in Renggli et al. (2022). We demonstrate $k$-NN's consistent performance across various transfer learning scenarios, along with its robustness w.r.t. $k$ and superior performance-runtime trade-off compared to all other metrics.

## 6 Conclusion

In this paper, we established the criteria for a reliable transferability metric, considering domain distance, target task complexity, and architectural differences. Assessing 23 existing metrics in this context, we observed that no existing single metric consistently performs well, or outperforms others. Metrics tailored for a specific transferability factor perform well when that factor is changed in isolation but struggle with varied factors, highlighting that a good metric should account for *all factors* to be useful. We proposed $k$-NN as a simple, robust alternative. $k$-NN proved effective in assessing the performance gains from transfer, and emerged as the best metric when considering all factors. It is cheap, optimization-free, and provides interpretable scores – making it an attractive choice for in practice.

**Acknowledgements.** This work was supported by the Wallenberg Autonomous Systems and Software Program (WASP) and MedTechLabs (MTL). We acknowledge the Berzelius computational resources provided by the Knut and Alice Wallenberg Foundation at the National Supercomputer Centre and the the computational resources provided by the National Academic Infrastructure for Supercomputing in Sweden (NAISS), partially funded by the Swedish Research Council through grant agreement no. 2022-06725.

## Impact Statement

This paper aims to advance the field of machine learning. One potential societal impact is improving resource efficiency in AI deployment, reducing the need for extensive computational experiments and lowering the environmental footprint of large-scale model training. While our focus is methodological, it can contribute to sustainable AI development by reducing resource requirements.

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

# Appendix for $k$-NN as a Simple and Effective Estimator of Transferability

**Appendix overview.** In this supplementary material, we provide more experimental details and results, as outlined below:

- more details regarding the implementation and datasets used in this work.

- the break-down of the results when domain and task complexity change together.

- the break-down of the results when only the domain changes.

- detailed results regarding when only the task complexity changes.

- additional results regarding the computational cost of different metrics, the effect of $k$ in $k$-NN, along with performance of metrics evaluated with different evaluation measures.

- example visualizations of feature representations

- concise mathematical formulations of representative transferability metrics

## A   More implementation details

For each dataset, either the official train/val/test splits were used, or we made the splits following Kornblith et al. (2019). Images were normalized and resized to $256 \times 256$, after which augmentations were applied: random color jittering, random horizontal flip and random cropping to $224 \times 224$ of the rescaled image. The Adam optimizer (Kingma & Ba, 2014) was used for CNNs and AdamW (Loshchilov & Hutter, 2017) for ViT-based architectures, and the training of models was done using PyTorch (Paszke et al., 2019). After a grid search, the pretrained and the randomly-initialized models were trained with a learning rate of $10^{-4}$ and $3 \times 10^{-4}$ respectively, following an initial warm-up for 1,000 iterations. During training, the learning rate was dropped by a factor of 10 whenever the training saturated until it reached a final learning rate of $10^{-6}$ or $3 \times 10^{-6}$ for pre-trained or randomly-initialized models respectively. The checkpoint with the highest validation performance was finally chosen for final evaluation.

## B   More details about the datasets

This section presents more details regarding the datasets used in this work. Starting with the source domains, IMAGENET (Deng et al., 2009; Russakovsky et al., 2015) contains around 1.2M training and 50,000 validation image from 1,000 classes. The PLACES365-Standard dataset (Zhou et al., 2017) used in this work contains 1.8M train and 36,500 validation images with 365 scene classes. iNat2017 (Van Horn et al., 2018) includes 675,170 train and validation images consisting of 5,089 classes of fine-grained species. NABirds (Van Horn et al., 2015) consists of 555 fine-grained bird classes and has 48,562 images in total.

Regarding the 12 target datasets utilized in this study, Table 3 provides the details of each, encompassing the number of images, number of classes, and their respective domain/task. Finally, Figure 6 illustrates an example of each target dataset.

## C   Additional results when domain and task complexity change

This section entails the detailed results for the scenario when both the target domain and task change from the source, corresponding to Figure 2 in the main paper. In this setup, a network is pre-trained on each of the four source datasets and subsequently transferred to each of the 12 target datasets. For each network,

Table 3: Summary of the downstream datasets used in this work.

| Dataset | Reference | Domain/task | Classes | Train size | Test size | Metric |
|---------|-----------|-------------|---------|-----------|-----------|--------|
| StanfordDogs | Khosla et al. (2011) | Fine-grained classification | 120 | 12,000 | 8,580 | Top-1 |
| Aircraft | Maji et al. (2013) | Fine-grained classification | 100 | 6,667 | 3,333 | Mean per class |
| NABirds | Van Horn et al. (2015) | Fine-grained classification | 555 | 23,929 | 24,633 | Top-1 |
| OxfordIII-Pet | Parkhi et al. (2012) | Fine-grained classification | 37 | 3,680 | 3,669 | Mean per class |
| CIFAR-10 | Krizhevsky et al. (2009) | Superordinate-level classification | 10 | 50,000 | 10,000 | Top-1 |
| CIFAR-100 | Krizhevsky et al. (2009) | Superordinate-level classification | 100 | 50,000 | 10,000 | Top-1 |
| Caltech-101 | Fei-Fei et al. (2004) | Superordinate-level classification | 101 | 3,030 | 5,647 | Mean per class |
| Caltech-256 | Griffin et al. (2007) | Superordinate-level classification | 257 | 15,420 | 15,187 | Mean per class |
| DTD | Cimpoi et al. (2014) | Texture classification | 47 | 3,760 | 1,880 | Top-1 |
| SUN397 | Xiao et al. (2010) | Scene classification | 397 | 19,850 | 19,850 | Top-1 |
| AID | Xia et al. (2017) | Aerial imagery | 30 | 5,000 | 5,000 | Top-1 |
| APTOS2019 | Karthik (2019) | Medical imaging | 5 | 3,113 | 549 | Qudratic Kappa |

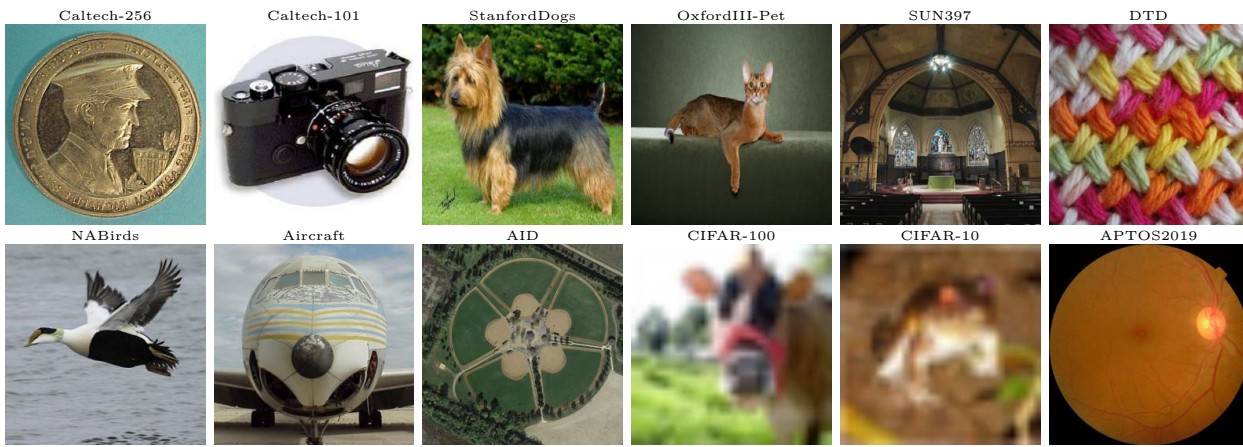

Figure 6: Overview of the downstream datasets used in this paper. These datasets cover superordinate-level object classification, fine-grained classification, scene recognition, texture classification, aerial imagery, and medical imaging.

we compute both the absolute and relative transfer performance, and then measure the correlation between these and the transferability metrics (including domain distance metrics). For each configuration, Tables 4 and 5 show the performance of different metrics in predicting absolute and relative transfer performance respectively. Figure 8 shows the inter-correlation of different transferability metrics. Figure 7 shows the same for domain distance metrics.

When predicting relative transfer performance (RTP), Table 5 shows that domain distance metrics are good predictors of the relative transfer performance (except for IMD).

However, the choice of the best distance metric is not consistent when the source dataset is different. Specifically, while EMD is the best domain distance metric when IMAGENET is the source domain ($\tau_w$: 0.68), IDS is the best when iNat2017 is the source ($\tau_w$: 0.69), and FID is the best when NABirds is the source ($\tau_w$: 0.74). This signifies that *no single definition of distance works best all the time* and the best distance metric could vary depending on the source domain. However, the differences in the distance definitions appears to be complementary. In fact, we see that "mean-dist", defined as the average of distance estimates, performs on average ($\tau_w$: 0.7) better than each individual domain distance metric, although at a higher cost.

# D  Additional results when only the domain changes

This section presents the detailed results regarding the scenario when only the domain distance changes – the target task and network architecture remain the same. This corresponds to Figure 3 in the main paper. In this setup, we use IMAGENET as the source domain and we fine-tune models on DomainNet (Peng et al., 2019). DomainNet comprises six distinct domains, each comprising the same set of 345 categories. As an

Table 4: Effectiveness of different metrics to predict *absolute transfer performance* when *target domain and task complexity change*. In this setup, a network is pre-trained on each of the four source datasets (shown in the top row) and subsequently transferred to each of the 12 target datasets. Quantitative correlations ($\tau_w$) of different metrics with absolute transfer performance in each configuration is shown below.

|  | ImageNet | iNat2017 | Places365 | NABirds | Average |
|---|---|---|---|---|---|
| TransRate | 0.33 | 0.24 | 0.12 | 0.03 | 0.18 |
| LogME | -0.38 | -0.48 | -0.55 | -0.48 | -0.47 |
| GBC | 0.60 | 0.72 | 0.67 | 0.70 | 0.67 |
| $\mathcal{N}$-LEEP | 0.49 | 0.52 | 0.53 | 0.47 | 0.50 |
| LEEP | 0.54 | 0.69 | 0.64 | 0.69 | 0.64 |
| H-Score | -0.40 | -0.46 | -0.42 | -0.44 | -0.43 |
| NCE | 0.37 | 0.67 | 0.62 | 0.71 | 0.59 |
| SFDA | 0.30 | 0.20 | 0.22 | 0.14 | 0.22 |
| NCTI | 0.39 | 0.66 | 0.59 | 0.59 | 0.56 |
| ETran | 0.47 | 0.55 | 0.59 | 0.49 | 0.52 |
| PACTran | 0.59 | 0.62 | 0.58 | 0.54 | 0.58 |
| PARC | 0.31 | 0.39 | 0.59 | 0.37 | 0.42 |
| OTCE | 0.65 | 0.33 | 0.66 | 0.69 | 0.58 |
| TMI | 0.54 | 0.40 | 0.67 | 0.62 | 0.56 |
| LFC | 0.43 | 0.57 | 0.43 | 0.43 | 0.46 |
| EMMS | 0.55 | 0.58 | 0.56 | 0.60 | 0.57 |
| NumC | 0.55 | 0.40 | 0.67 | 0.58 | 0.55 |
| FID | -0.25 | -0.24 | -0.09 | -0.21 | -0.20 |
| KID | -0.13 | -0.25 | -0.23 | -0.22 | -0.21 |
| EMD | -0.33 | -0.26 | -0.29 | -0.35 | -0.31 |
| IDS | -0.22 | -0.28 | -0.33 | -0.19 | -0.26 |
| IMD | 0.10 | -0.23 | -0.21 | -0.31 | -0.16 |
| mean-dist | -0.22 | -0.37 | -0.29 | -0.33 | -0.30 |
| $k$-NN | 0.54 | 0.62 | 0.64 | 0.61 | 0.60 |

Table 5: Effectiveness of different metrics to predict *relative transfer performance* when *target domain and task complexity change*. In this setup, a network is pre-trained on each of the four source datasets (shown in the top row) and subsequently transferred to each of the 12 target datasets. Quantitative correlations ($\tau_w$) of different metrics with relative transfer performance in each configuration is shown below.

|  | ImageNet | iNat2017 | Places365 | NABirds | Average |
|---|---|---|---|---|---|
| TransRate | 0.36 | 0.29 | 0.27 | 0.51 | 0.36 |
| LogME | -0.14 | -0.18 | -0.20 | -0.01 | -0.13 |
| GBC | 0.71 | 0.68 | 0.66 | 0.70 | 0.69 |
| $\mathcal{N}$-LEEP | 0.46 | 0.47 | 0.18 | 0.54 | 0.41 |
| LEEP | 0.57 | 0.67 | 0.33 | 0.64 | 0.55 |
| H-Score | 0.32 | 0.35 | 0.17 | 0.52 | 0.34 |
| NCE | 0.60 | 0.67 | 0.27 | 0.74 | 0.57 |
| SFDA | 0.00 | -0.11 | -0.27 | -0.14 | -0.13 |
| NCTI | 0.21 | -0.03 | -0.18 | 0.01 | 0.00 |
| ETran | 0.69 | 0.66 | 0.49 | 0.75 | 0.65 |
| PACTran | 0.75 | 0.72 | 0.71 | 0.77 | 0.74 |
| PARC | 0.43 | 0.51 | 0.39 | 0.52 | 0.46 |
| OTCE | 0.73 | 0.69 | 0.68 | 0.70 | 0.7 |
| TMI | -0.05 | 0.54 | 0.28 | 0.10 | 0.22 |
| LFC | -0.03 | 0.02 | -0.25 | 0.12 | -0.04 |
| EMMS | 0.52 | 0.49 | 0.37 | 0.53 | 0.48 |
| NumC | -0.67 | -0.66 | -0.59 | -0.59 | -0.63 |
| FID | 0.59 | 0.64 | 0.39 | 0.74 | 0.59 |
| KID | 0.57 | 0.61 | 0.51 | 0.63 | 0.58 |
| EMD | 0.68 | 0.57 | 0.43 | 0.67 | 0.59 |
| IDS | 0.65 | 0.69 | 0.65 | 0.60 | 0.65 |
| IMD | 0.23 | 0.20 | 0.11 | 0.19 | 0.18 |
| mean-dist | 0.69 | 0.71 | 0.59 | 0.81 | 0.70 |
| $k$-NN | 0.89 | 0.95 | 0.84 | 0.93 | 0.90 |

additional experiment, we set each individual domain, within DomainNet, as the source and use as targets the remaining domains. For each configuration, Tables 6 and 7 show how well different metrics perform when predicting absolute and relative transfer performance respectively.

Considering absolute transfer performance, Table 6 demonstrates that $k$-NN ($\tau_w$: 0.65) appears to be the best metric, followed by LogME ($\tau_w$: 0.63) and H-score ($\tau_w$: 0.63). Unlike the previous setting, GBC ($\tau_w$: 0.36) is no longer a good metric in this setting. When predicting relative transfer performance, Table 7 shows

**ImageNet**

|     | FID | KID | EMD | IDS | IMD |
|-----|-----|-----|-----|-----|-----|
| FID | 1.0 | 0.94 | 0.91 | 0.35 | 0.15 |
| KID | 0.94 | 1.0 | 0.84 | 0.34 | 0.2 |
| EMD | 0.91 | 0.84 | 1.0 | 0.41 | 0.2 |
| IDS | 0.35 | 0.34 | 0.41 | 1.0 | 0.08 |
| IMD | 0.15 | 0.2 | 0.2 | 0.08 | 1.0 |

**iNat2017**

|     | FID | KID | EMD | IDS | IMD |
|-----|-----|-----|-----|-----|-----|
| FID | 1.0 | 0.97 | 0.88 | 0.42 | 0.14 |
| KID | 0.97 | 1.0 | 0.9 | 0.38 | 0.12 |
| EMD | 0.88 | 0.9 | 1.0 | 0.28 | 0.21 |
| IDS | 0.42 | 0.38 | 0.28 | 1.0 | 0.06 |
| IMD | 0.14 | 0.12 | 0.21 | 0.06 | 1.0 |

**Places365**

|     | FID | KID | EMD | IDS | IMD |
|-----|-----|-----|-----|-----|-----|
| FID | 1.0 | 0.87 | 0.89 | 0.47 | 0.12 |
| KID | 0.87 | 1.0 | 0.83 | 0.58 | 0.07 |
| EMD | 0.89 | 0.83 | 1.0 | 0.47 | 0.18 |
| IDS | 0.47 | 0.58 | 0.47 | 1.0 | 0.25 |
| IMD | 0.12 | 0.07 | 0.18 | 0.25 | 1.0 |

**NABirds**

|     | FID | KID | EMD | IDS | IMD |
|-----|-----|-----|-----|-----|-----|
| FID | 1.0 | 0.78 | 0.73 | 0.6 | 0.18 |
| KID | 0.78 | 1.0 | 0.76 | 0.28 | 0.1 |
| EMD | 0.73 | 0.76 | 1.0 | 0.29 | 0.22 |
| IDS | 0.6 | 0.28 | 0.29 | 1.0 | -0.0 |
| IMD | 0.18 | 0.1 | 0.22 | -0.0 | 1.0 |

Figure 7: *Cross correlations of different domain distance definitions.* The distances are calculated from each source (shown on top of each panel) to 12 target datasets.

Table 6: Effectiveness of different metrics to predict *absolute transfer performance* when *only the target domain changes.* In this setup, IMAGENET is chosen as the source and fine-tuning is done on DomainNet (Peng et al., 2019). Additionally, we set each individual domain within DomainNet as the source (shown in the top row) and use the remaining domains as targets, resulting in a total of seven configurations. Quantitative correlations ($\tau_w$) of different metrics with absolute transfer performance in each configuration is shown below.

|  | ImageNet | Clip. | Info. | Paint. | Quick. | Real | Sketch | Average |
|---|---|---|---|---|---|---|---|---|
| TransRate | -0.10 | -0.40 | 0.35 | 0.00 | -0.15 | -0.05 | 0.03 | -0.05 |
| LogME | 0.87 | 0.74 | 0.11 | 0.74 | 0.51 | 0.65 | 0.81 | 0.63 |
| GBC | 0.41 | 0.22 | 0.29 | -0.37 | 0.25 | 0.79 | 0.90 | 0.36 |
| $\mathcal{N}$-LEEP | 0.36 | 0.39 | 0.20 | 0.45 | 0.33 | 0.38 | 0.67 | 0.40 |
| LEEP | 0.33 | 0.20 | 0.16 | 0.45 | 0.68 | 0.31 | 0.67 | 0.40 |
| H-Score | 0.72 | 0.53 | 0.30 | 0.67 | 0.74 | 0.63 | 0.81 | 0.63 |
| NCE | 0.33 | 0.20 | 0.20 | 0.45 | 0.68 | 0.31 | 0.67 | 0.41 |
| SFDA | 0.60 | 0.37 | 0.54 | 0.61 | 0.34 | 0.26 | 0.62 | 0.48 |
| NCTI | 0.70 | 0.61 | 0.05 | 0.26 | 0.35 | 0.78 | 0.29 | 0.43 |
| ETran | 0.70 | 0.61 | 0.34 | 0.26 | 0.35 | 0.60 | 0.29 | 0.45 |
| PACTran | 0.05 | 0.10 | -0.28 | 0.11 | 0.52 | 0.34 | 0.55 | 0.20 |
| PARC | 0.28 | -0.11 | 0.40 | 0.76 | 0.76 | 0.87 | 0.40 | 0.48 |
| OTCE | 0.14 | 0.26 | -0.16 | 0.43 | 0.46 | 0.23 | 0.53 | 0.27 |
| TMI | -0.19 | -0.46 | -0.34 | -0.45 | 0.32 | -0.47 | -0.55 | -0.31 |
| LFC | 0.66 | 0.59 | 0.16 | 0.67 | 0.79 | 0.65 | 0.82 | 0.62 |
| EMMS | 0.61 | 0.59 | 0.20 | 0.60 | 0.74 | 0.46 | 0.79 | 0.57 |
| FID | 0.19 | -0.02 | 0.08 | 0.45 | 0.02 | 0.15 | 0.24 | 0.16 |
| KID | 0.37 | -0.03 | 0.06 | 0.45 | 0.50 | 0.21 | 0.24 | 0.26 |
| EMD | 0.19 | -0.19 | -0.16 | 0.32 | -0.12 | 0.21 | 0.17 | 0.06 |
| IDS | -0.17 | -0.38 | 0.32 | 0.01 | 0.15 | -0.09 | -0.18 | -0.05 |
| IMD | 0.26 | -0.42 | -0.54 | -0.41 | 0.63 | 0.26 | -0.34 | -0.08 |
| mean-dist | 0.13 | -0.21 | 0.05 | 0.01 | 0.29 | 0.14 | 0.10 | 0.07 |
| $k$-NN | 0.64 | 0.55 | 0.20 | 0.82 | 0.79 | 0.71 | 0.84 | 0.65 |

that $k$-NN ($\tau_w$: 0.59) is again the strongest metric on average, followed by mean-dist ($\tau_w$: 0.55) and EMD ($\tau_w$: 0.54). In contrast to the previous setting, PACTran ($\tau_w$: -0.01) shows negative correlation on average. Overall, these result further confirm that the choice of the best transferability metric is not consistent and changes depending on the experimental setup. Further, it suggests that $k$-NN matches or outperforms the best metric in each setting.

## E   Additional results when only the task complexity changes

This section presents the detailed results corresponding to the case of changing only the target task – the domain and architecture remain the same. This corresponds to Figure 4 in the main paper. Using IMAGENET as the source, we apply transfer learning to subsets of NABirds, SUN397, and Caltech-256. More specifically, we vary the task complexity by varying the number of classes (as a proxy) in accordance with Deng et al. (2010); Konuk & Smith (2019). In detail, we sequentially remove classes from the target dataset to create target tasks of varying complexity. For each configuration, Tables 8 and 9 display the performance of different metrics when predicting absolute and relative transfer performance respectively.

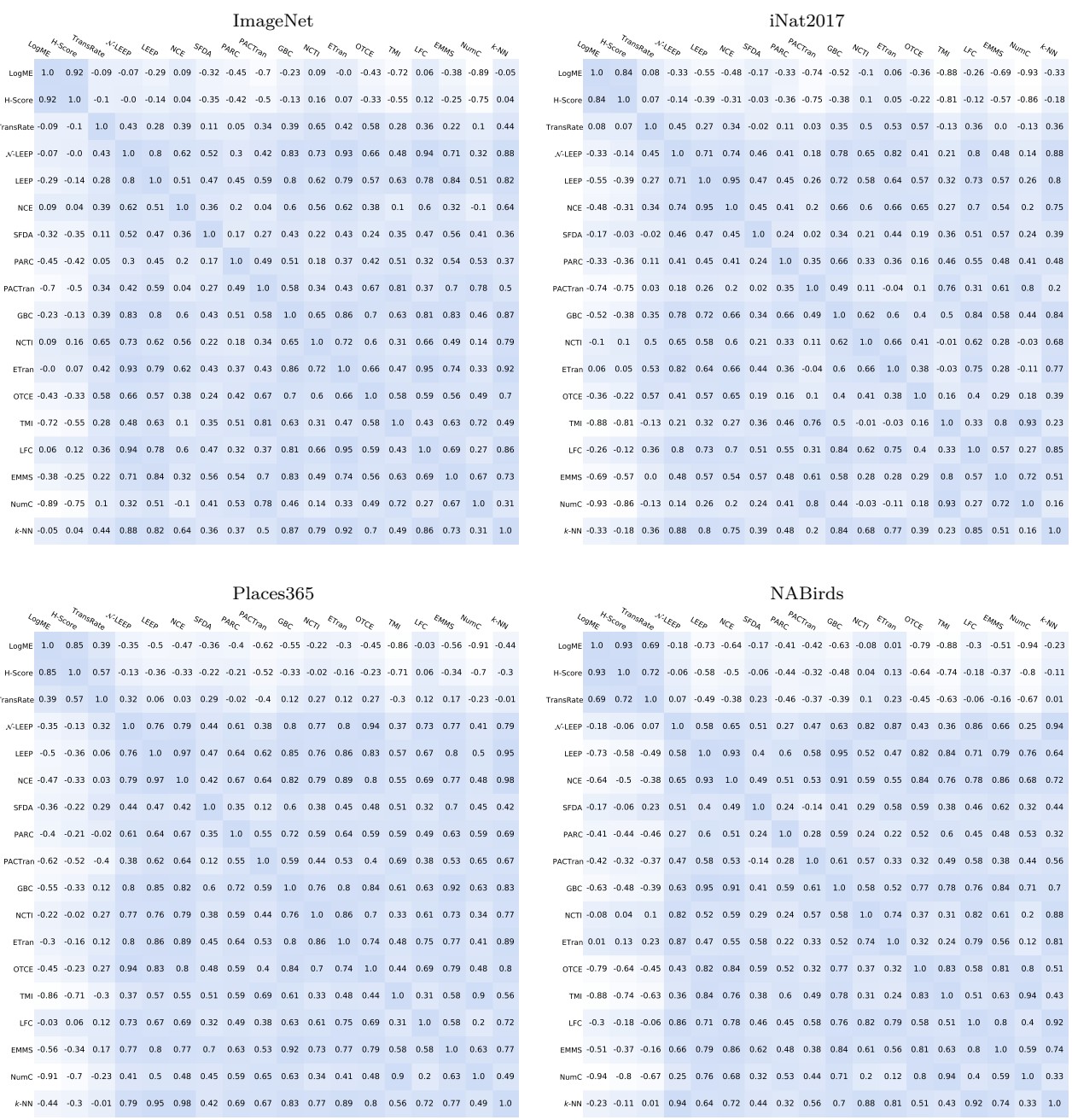

Figure 8: *Inter-correlations of different transferability metrics.* The transferability scores are calculated from each source (shown on top of each panel) to all the target datasets. Evidently, some metrics demonstrate stronger correlations than others. For instance, LogME and H-score display a high correlation with each other but a low correlation with the other metrics. GBC, $\mathcal{N}$-LEEP, and *k*-NN exhibit strong correlations with one another.

When predicting absolute transfer performance, NCE, LEEP, $\mathcal{N}$-LEEP, GBC, ETran, and PACTran emerge as the top performers ($\tau_w$: 0.98 on average). *k*-NN proves to be an effective estimator of transferability in this setting, and performs on par with the best transferability metrics ($\tau_w$: 0.98 on average).

When predicting relative transfer performance, NCE. LEEP, GBC, NCTI, ETran, and PACTran perform best among the transferability metrics ($\tau_w$: 0.93 on average), and FID and mean-dist show similar performance

Table 7: Effectiveness of different metrics to predict *relative transfer performance* when *only the target domain changes.* In this setup, IMAGENET is chosen as the source and fine-tuning is done on DomainNet (Peng et al., 2019). Additionally, we set each individual domain within DomainNet as the source (shown in the top row) and use the remaining domains as targets, resulting in a total of seven configurations. Quantitative correlations ($\tau_w$) of different metrics with relative transfer performance in each configuration is shown below.

| | ImageNet | Clip. | Info. | Paint. | Quick. | Real | Sketch | Average |
|---|---|---|---|---|---|---|---|---|
| TransRate | 0.20 | 0.21 | 0.28 | 0.10 | 0.08 | 0.14 | 0.11 | 0.16 |
| LogME | 0.44 | 0.31 | 0.84 | 0.17 | 0.71 | 0.32 | 0.31 | 0.44 |
| GBC | 0.08 | 0.44 | 0.43 | 0.52 | -0.19 | 0.08 | 0.39 | 0.25 |
| $\mathcal{N}$-LEEP | -0.10 | -0.43 | -0.14 | -0.04 | 0.19 | -0.03 | -0.43 | -0.14 |
| LEEP | 0.45 | -0.07 | 0.49 | 0.46 | 0.34 | 0.54 | -0.09 | 0.30 |
| H-Score | 0.26 | 0.16 | 0.41 | 0.34 | 0.31 | 0.47 | 0.24 | 0.31 |
| NCE | 0.24 | -0.15 | 0.41 | 0.34 | 0.47 | 0.42 | -0.09 | 0.23 |
| SFDA | -0.25 | -0.24 | -0.09 | -0.14 | 0.24 | -0.18 | -0.39 | -0.15 |
| NCTI | 0.02 | -0.53 | 0.38 | 0.33 | 0.28 | -0.09 | 0.13 | 0.07 |
| ETran | -0.14 | -0.20 | -0.04 | 0.51 | 0.57 | 0.18 | 0.46 | 0.19 |
| PACTran | 0.17 | 0.05 | -0.56 | 0.38 | -0.04 | 0.04 | -0.08 | -0.01 |
| PARC | 0.15 | 0.19 | -0.46 | -0.47 | 0.45 | -0.32 | 0.19 | -0.04 |
| OTCE | -0.48 | -0.66 | -0.50 | -0.54 | 0.14 | -0.52 | -0.67 | -0.46 |
| TMI | 0.50 | 0.41 | 0.49 | 0.47 | 0.24 | 0.58 | 0.27 | 0.42 |
| LFC | 0.24 | -0.13 | 0.49 | 0.22 | 0.39 | 0.28 | -0.07 | 0.2 |
| EMMS | 0.41 | 0.55 | 0.49 | 0.53 | 0.40 | 0.48 | 0.59 | 0.49 |
| FID | 0.30 | 0.49 | 0.64 | 0.46 | 0.45 | 0.76 | 0.55 | 0.52 |
| KID | 0.42 | 0.51 | 0.64 | 0.46 | 0.44 | 0.72 | 0.55 | 0.53 |
| EMD | 0.49 | 0.60 | 0.82 | 0.40 | 0.15 | 0.72 | 0.60 | 0.54 |
| IDS | 0.45 | 0.46 | 0.32 | 0.10 | 0.11 | 0.28 | 0.10 | 0.26 |
| IMD | 0.30 | 0.50 | 0.66 | 0.71 | 0.07 | 0.13 | 0.73 | 0.44 |
| mean-dist | 0.49 | 0.58 | 0.67 | 0.56 | 0.33 | 0.61 | 0.59 | 0.55 |
| $k$-NN | 0.58 | 0.62 | 0.61 | 0.44 | 0.51 | 0.72 | 0.66 | 0.59 |

Table 8: Effectiveness of different metrics in predicting *absolute transfer performance* when *only the target task complexity changes.* In this setup, IMAGENET is used as the source, and fine-tuning is done on subsets of NABirds, SUN397, and Caltech-256, where each subset consists of varying number of classes (see main text for details). Quantitative correlations ($\tau_w$) of different metrics with absolute transfer performance in each configuration is shown below.

| | NABirds | SUN397 | Caltech-256 | Average |
|---|---|---|---|---|
| TransRate | 0.58 | 0.50 | 0.64 | 0.57 |
| LogME | -0.35 | -0.23 | -0.14 | -0.24 |
| GBC | 1.00 | 0.97 | 0.96 | 0.98 |
| $\mathcal{N}$-LEEP | 1.00 | 0.97 | 0.96 | 0.98 |
| LEEP | 1.00 | 0.97 | 0.96 | 0.98 |
| H-Score | -0.69 | -0.83 | -0.61 | -0.71 |
| NCE | 1.00 | 0.97 | 0.96 | 0.98 |
| SFDA | 0.97 | 0.94 | 0.95 | 0.95 |
| NCTI | 0.92 | 0.97 | 0.61 | 0.83 |
| ETran | 1.00 | 0.97 | 0.96 | 0.98 |
| PACTran | 1.00 | 0.97 | 0.96 | 0.98 |
| PARC | 0.88 | 0.82 | 0.83 | 0.84 |
| OTCE | 0.81 | 0.92 | 0.92 | 0.88 |
| TMI | 1.00 | 0.97 | 0.96 | 0.98 |
| LFC | 1.00 | 0.97 | 0.97 | 0.98 |
| EMMS | 0.99 | 0.98 | 0.97 | 0.98 |
| NumC | 1.00 | 0.97 | 0.96 | 0.98 |
| FID | -0.97 | -0.91 | -0.88 | -0.92 |
| KID | -0.79 | -0.68 | -0.88 | -0.78 |
| EMD | -1.00 | -0.94 | -0.88 | -0.94 |
| IDS | 0.54 | 0.30 | 0.29 | 0.38 |
| IMD | 0.85 | 0.76 | -0.48 | 0.38 |
| mean-dist | -1.00 | -0.91 | -0.88 | -0.93 |
| $k$-NN | 1.00 | 0.97 | 0.96 | 0.98 |

($\tau_w$: 0.93 on average). Notably, H-score and LogME show negative correlation when predicting both absolute and relative transfer performance in this setup, indicating their inability to account for task complexity. $k$-NN shows strong performance ($\tau_w$: 0.93 on average), on par with the best transferability and domain distance metrics. Overall, these results show that $k$-NN can successfully account for task complexity as well.

Table 9: Effectiveness of different metrics for predicting *relative transfer performance* when *only the target task complexity changes*. In this setup, IMAGENET is used as the source, and fine-tuning is done on subsets of NABirds, SUN397, and Caltech-256, where each subset consists of varying number of classes (see text for details). Quantitative correlations ($\tau_w$) of different metrics with relative transfer performance in each configuration is shown below.

| | NABirds | SUN397 | Caltech-256 | Average |
|---|---|---|---|---|
| TransRate | -0.63 | -0.68 | -0.77 | -0.69 |
| LogME | -0.42 | -0.35 | -0.18 | -0.32 |
| GBC | 0.89 | 0.93 | 0.97 | 0.93 |
| $\mathcal{N}$-LEEP | 0.71 | 0.86 | 0.97 | 0.85 |
| LEEP | 0.89 | 0.93 | 0.97 | 0.93 |
| H-Score | 0.11 | -0.01 | -0.14 | -0.01 |
| NCE | 0.89 | 0.93 | 0.97 | 0.93 |
| SFDA | 0.69 | 0.44 | 0.83 | 0.65 |
| NCTI | 0.89 | 0.94 | 0.96 | 0.93 |
| ETran | 0.89 | 0.94 | 0.97 | 0.93 |
| PACTran | 0.89 | 0.93 | 0.97 | 0.93 |
| PARC | 0.68 | 0.60 | 0.53 | 0.6 |
| OTCE | 0.88 | 0.94 | 0.97 | 0.93 |
| TMI | 0.63 | 0.40 | 0.01 | 0.35 |
| LFC | -0.75 | -0.81 | -0.60 | -0.72 |
| EMMS | 0.84 | 0.89 | 0.94 | 0.89 |
| NumC | -0.77 | -0.85 | -0.94 | -0.85 |
| FID | 0.89 | 0.93 | 0.97 | 0.93 |
| KID | 0.86 | 0.88 | 0.97 | 0.9 |
| EMD | 0.89 | 0.90 | 0.97 | 0.92 |
| IDS | -0.58 | -0.55 | -0.47 | -0.53 |
| IMD | -0.69 | -0.75 | 0.27 | -0.39 |
| mean-dist | 0.89 | 0.93 | 0.97 | 0.93 |
| $k$-NN | 0.89 | 0.93 | 0.97 | 0.93 |

All together, these findings highlight the capability of $k$-NN as a robust transferability metric, which can successfully account for crucial factors of transferability better than all the current transferability metrics from the literature.

## F   Further ablation studies

In this section, we provide additional results when using evaluation measures, such as Kendall's $\tau$ (Kendall, 1938), Pearson's $\rho$ (Pearson, 1895), and Rel@1 (Li et al., 2021) (Figure 10). We further include ablation studies on the transferability metrics' total runtime (Figure 9a) and robustness of $k$-NN to the choice of $k$ (Figure 9b).

## G   A theoretical formulation for transferability measures

For a more unified understanding, we can frame transferability measures in terms of probably approximately correct (PAC) learning. Our goal is not to derive tight generalization bounds for the measures in the literature but provide a theoretical framework to formulate the problem for better insights.

Supervised learning aims to approximate the true conditional distribution, $P_T$, with a learned distribution $\hat{P}_T$, usually by minimizing the KL-divergence between them, $D_{KL}(P_T||\hat{P}_T)$, whereas in transfer learning we minimize the divergence, $D_{KL}(P_T||\hat{P}_{T|S})$, starting from a source-pretrained model, $\hat{P}_{T|S}$ . Here, we slightly abuse the notation with $S$ indicating a pretrained model and not just the pretraining dataset. Unlike NCE (Tran et al., 2019) we are not limiting the problem to a setting where the input instances are shared and only the conditional distributions differ across tasks.

PAC interpretation of a transfer learning task for small $\epsilon$ and $\delta$ can be given as,

$$P(|err(\hat{P}_{T|S}) - err_B - \inf_{h \in \mathbb{S}} err(h)| \leq \epsilon) \geq 1 - \delta \tag{1}$$

where $err_B$ is the irreducible Bayes error of the target distribution and $\mathbb{S}$ indicates the hypothesis class of target models initialized with a *given* source-pretrained model. Note that $err_B$ is sometimes omitted in PAC formulation since it is generally not feasible to achieve, *i.e.* $D_{KL}(P_T||\hat{P}_{T|S}) \neq 0$. We include it here to highlight the conceptual distinction between the goals of transfer learning and the notion of transferability, as discussed below.

In the strictest sense PAC addresses *all possible target distributions* and the infimum operator is blind to the training method thus different initializations has no impact on the hypothesis class. PAC framing simply states the aim of transfer learning as the estimation of the true data distribution, $P_T$, *i.e.* approximation of Bayes error, $err_B$, ideally without using many samples (or low sample complexity). Instead, the concept of non-trivial transfer introduced in Galanti et al. (2016), together with empirical observations that pretrained models require fewer labeled examples to perform well (Azizpour et al., 2015; Raghu et al., 2019; Matsoukas et al., 2022), support the hypothesis that for the subset of target distributions relevant to computer vision, a randomly initialized model typically requires more samples to achieve error rates comparable to a well-initialized model using pretrained weights.

The aim of transferability methods is *to predict how well a tuned model will perform*, *i.e.* to provide an approximation of a fine-tuned model's error, ideally without training said model. In other words, instead of estimating the true distribution, *e.g.* using empirical risk minimization based standard deep learning methods, such that the learned model approximates the error of a (hypothetical) Bayes optimal model, we aim to estimate a given fine-tuned model's error, $err(\hat{P}_{T|S})$. Following a similar PAC formulation as transfer learning,

$$P(|err(\hat{m}_T(S)) - err(\hat{P}_{T|S}) - \inf_{m \in \mathbb{M}} m_T(S)| \leq \epsilon) \geq 1 - \delta \tag{2}$$

where $m_T(S)$ indicates the transferability error from a source domain $S$ to a target task $T$ and $\mathbb{M}$ is the model family used for estimating transferability. It would be safe to assume that $\mathbb{M}$ should be a simpler hypothesis class than $\mathbb{S}$.

This framing formalizes the intuition behind the standard empirical evaluation approaches, *i.e.* to calculate a transferability measures' correlation (or anti-correlation) to downstream accuracy. After all, the best transferability measure is the transfer learned model itself. Unfortunately, to the best of our knowledge, there are no tight generalization bounds proposed in the literature for this setting, and even in a limited problem setting (Tran et al., 2019) bounds require strong assumptions.

Regardless, even without theoretical bounds, PAC framing of transferability can inform our understanding of various approaches in terms of how tightly they can approximate the transferred model. Below, we compare and contrast $k$-NN with its most similar metrics, GBC and $\mathcal{N}$-LEEP, to understand why it may offer a better estimate of transferability. We examine these metrics through the lens of estimated error. Transferability is naturally inversely related to the estimated error.

- **GBC** aims to measure class separability - an indicator of classifier performance. They propose fitting Gaussians to the target classes in the source embedding space. They measure the Bhattacharyya distance between these target classes as a measure of their separability, using this an indicator for how transferable the source model is. This separability can also be understood as the error rate of a (Gaussian) naive Bayes classifier. For a simplified binary problem, the error of a naive Bayes classifier is given by,

$$err_G = P(Y=1) \int_{-\infty}^{B} p(x|Y=1)dx + P(Y=0) \int_{B}^{+\infty} p(x|Y=0)dx$$

  where the decision boundary $B$ directly controls the error rate and it is easy to see that low Bhattacharyya distance between two classes leads $B$ to be determined with more entangled classes, yielding a high error rate.

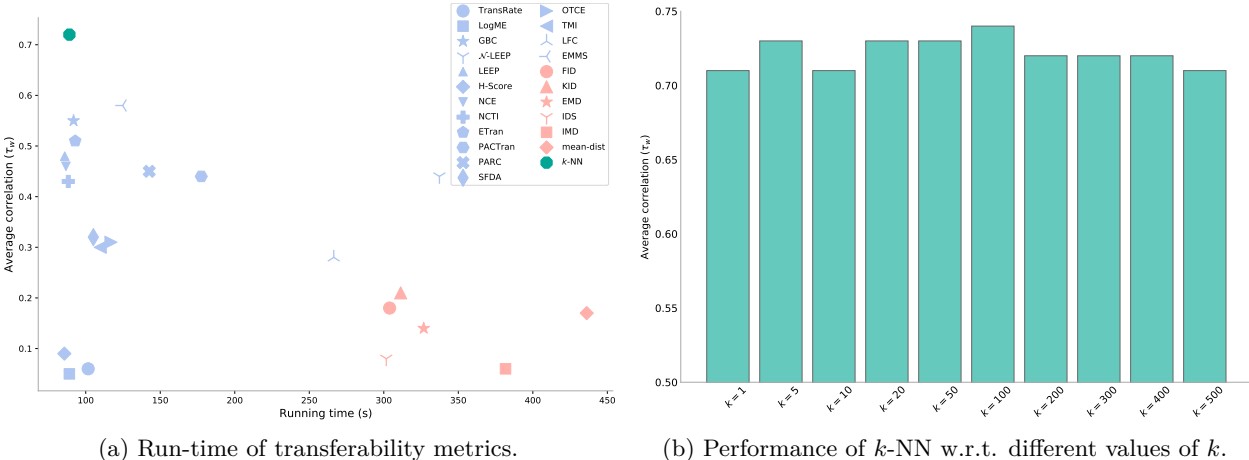

(a) Run-time of transferability metrics.

(b) Performance of $k$-NN w.r.t. different values of $k$.

Figure 9: (Left) comparison of run-time against performance for different metrics. (Right) robustness of $k$-NN to the choice of $k$.

- $\mathcal{N}$-**LEEP** is explicitly defined as a classifier. This method is a bit more relaxed than GBC in the sense that the $\mathcal{N}$-LEEP measure, *i.e.* log error rate of this classifier, is based not on a single Gaussian per class but a Gaussian mixture model in the source embedding space. We can intuitively understand $\mathcal{N}$-LEEP as a more expressive hypothesis family than naive Bayes classifiers.

- $k$-**NN** is our proposed method, which eliminates the need for Gaussian or mixture of Gaussian assumptions. Instead, it directly leverages the inherent distribution of features extracted from the source model for classification, making it inherently more flexible and expressive.

- **Fine-tuning** is the gold-standard classifier but is computationally expensive and impractical for real-world use. It is the most flexible approach, as it enables network weights to adjust to the target distribution.

As an additional advantage of using an expressive yet computationally efficient non-parametric classifier like $k$-NN, we note that $k$-NN can be more robust to outliers compared to Gaussian or GMM models. Notably, $k$-NN serves as a Bayes-optimal classifier for uniform distributions, whereas (Gaussian) naive Bayes is naturally optimal for normal distributions [1]. However, drawing definitive conclusions based on these theoretical properties may not ne entirely meaningful, as neither assumption can be reliably validated in the high-dimensional representation spaces of deep learning models. Instead, empirical observations should guide further research in this area.

## H  Feature visualization

In this section, we present t-SNE (Van der Maaten & Hinton, 2008) visualizations of feature representations extracted from the penultimate layer of a ResNet-50 model $P$ pre-trained on ImageNet, evaluated across several downstream tasks. The results, shown in Figure 10, illustrate that higher $k$-NN transferability estimates $est(P)$ indicate higher transfer performance $perf(P)$.

## I  Mathematical formulations

In this section, we present concise mathematical formulations of several representative metrics of transferability considered in this study.

**NCE (Tran et al., 2019)**    assumes the source and target datasets share the same input examples. It treats the source and target labels as random variables and uses their conditional entropy to quantify how much

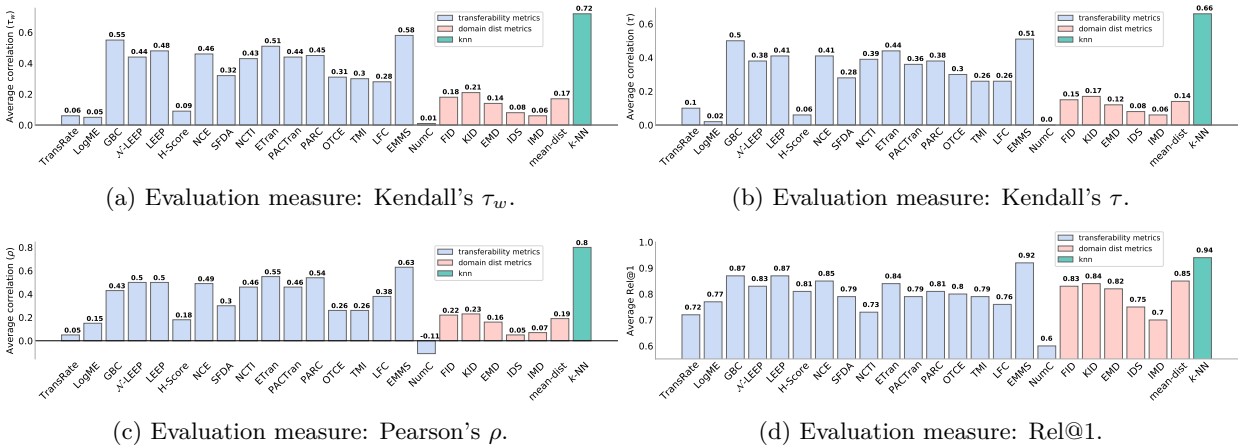

(a) Evaluation measure: Kendall's $\tau_w$.

(b) Evaluation measure: Kendall's $\tau$.

(c) Evaluation measure: Pearson's $\rho$.

(d) Evaluation measure: Rel@1.

Figure 10: Transferability metrics evaluated using common correlation coefficient measures.

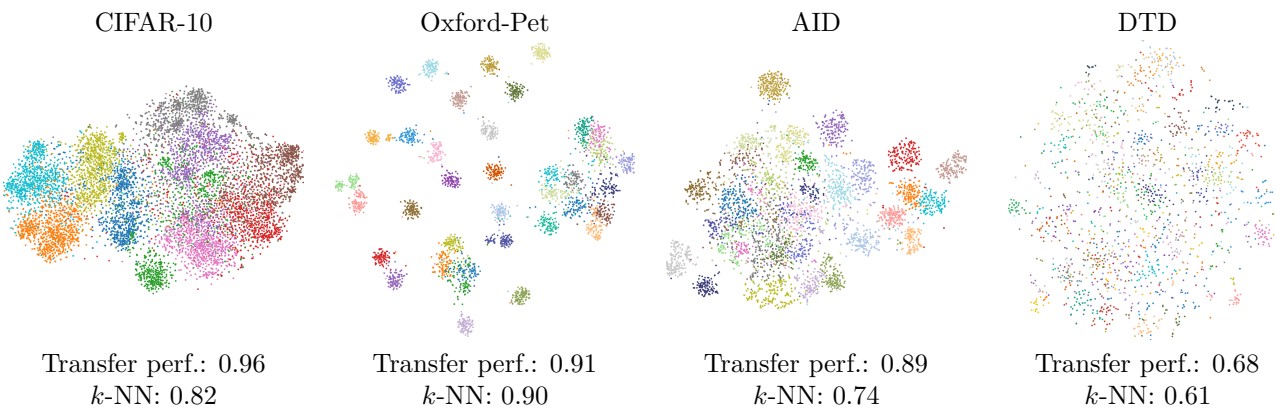

| CIFAR-10 | Oxford-Pet | AID | DTD |
| --- | --- | --- | --- |
| Transfer perf.: 0.96 | Transfer perf.: 0.91 | Transfer perf.: 0.89 | Transfer perf.: 0.68 |
| $k$-NN: 0.82 | $k$-NN: 0.90 | $k$-NN: 0.74 | $k$-NN: 0.61 |

Table 10: t-SNE visualizations of feature representations from the penultimate layer of a ResNet-50 model $P$ pre-trained on ImageNet, evaluated on various downstream tasks. Each color represents a distinct class. For each task, we report the $k$-NN transferability estimate $est(P)$ prior to fine-tuning, alongside the absolute transfer performance $perf(P)$ after fine-tuning. Tasks with higher $k$-NN estimates have better transfer performance.

information the source task label provides about the target task label. It considers a single sequence of training samples $X = (x_1, x_2, \ldots, x_n) \in \mathcal{X}^n$, along with two corresponding label sequences $Y = (y_1, y_2, \ldots, y_n) \in \mathcal{Y}^n$ and $Z = (z_1, z_2, \ldots, z_n) \in \mathcal{Z}^n$, where $y_i$ and $z_i$ are labels assigned to $x_i$ under two different tasks: the source task $T_Z = (\mathcal{X}, \mathcal{Z})$ and the target task $T_Y = (\mathcal{X}, \mathcal{Y})$. Here, $\mathcal{X}$ denotes the domain of the values of $X$, while $\mathcal{Y}$ and $\mathcal{Z}$ represent the sets of unique labels in $Y$ and $Z$, respectively.

Transferability between the two tasks is estimated by computing the conditional entropy (CE) between the two label sequences. First, from the label sequences $Y$ and $Z$, the empirical joint distribution $\hat{P}(y, z)$ for all $(y, z) \in \mathcal{Y} \times \mathcal{Z}$ is computed as:

$$\hat{P}(y, z) = \frac{1}{n} \left| \{ i : y_i = y \text{ and } z_i = z \} \right|,$$

where $|\cdot|$ denotes the cardinality of the set. Then, the definition of CE between two random variables (Cover, 1999) is employed to compute the CE between the two label sequences $Y$ and $Z$:

$$H(Y \mid Z) = - \sum_{y \in \mathcal{Y}} \sum_{z \in \mathcal{Z}} \hat{P}(y, z) \log \frac{\hat{P}(y, z)}{\hat{P}(z)},$$

where $\hat{P}(z)$ is the empirical marginal distribution over $Z$, given by:

$$\hat{P}(z) = \sum_{y \in \mathcal{Y}} \hat{P}(y, z) = \frac{1}{n} \left| \{i : z_i = z\} \right|.$$

Since lower conditional entropy indicates better transferability, the negative conditional entropy, $-H(Y \mid Z)$, is used as the final transferability metric for predicting transfer performance.

**Log Expected Empirical Prediction (LEEP)** Nguyen et al. (2020) estimate transferability of a source model $\theta$ pre-trained on a classification task with output label space $\mathcal{Z}$ to a target classification task with dataset $\mathcal{D} = \{(x_1, y_1), (x_2, y_2), \ldots, (x_n, y_n)\}$, where $x_i \in \mathcal{X}$ and $y_i \in \mathcal{Y}$. For each target input $x_i$, the model outputs a softmax distribution $\theta(x_i)$, where $\theta(x_i)_z$ denotes the probability of assigning $x_i$ to class $z \in \mathcal{Z}$. First, the empirical joint distribution between source and target labels is computed as:

$$\hat{P}(y, z) = \frac{1}{n} \sum_{i:y_i=y} \theta(x_i)_z,$$

where summation $\sum_{i:y_i=y}$ indicates that we sum over all indices $i \in \{1, 2, \ldots, n\}$ for which the target label $y_i$ equals $y$. This is then used to compute the empirical conditional distribution:

$$\hat{P}(y \mid z) = \frac{\hat{P}(y, z)}{\sum_{y' \in \mathcal{Y}} \hat{P}(y', z)}.$$

For any input instance $x \in \mathcal{X}$, a label $y$ is predicted by first sampling $z \sim \theta(x)$, then $y \sim \hat{P}(y|z)$, which is equivalent to predicting $y$ using the distribution:

$$p(y \mid x; \theta, \mathcal{D}) = \sum_{z \in \mathcal{Z}} \hat{P}(y \mid z)\, \theta(x)_z.$$

The final LEEP score, representing the average log-likelihood of the target labels under this expected predictor, is defined as:

$$\text{LEEP}(\theta, \mathcal{D}) = \frac{1}{n} \sum_{i=1}^{n} \log \left( \sum_{z \in \mathcal{Z}} \hat{P}(y_i \mid z)\, \theta(x_i)_z \right).$$

$\mathcal{N}$-**LEEP** Li et al. (2021) extend the original LEEP score to enable estimating the transferability of models pre-trained via unsupervised tasks, without requiring a source classification head. Given a target dataset $\mathcal{D} = \{(x_1, y_1), (x_2, y_2), \ldots, (x_n, y_n)\}$, they extract the feature representation $s_i$ of input $x_i$ using the pre-trained model. They then fit a Gaussian Mixture Model (GMM) on the set $\{s_i\}_{i=1}^{n}$:

$$P(s) = \sum_{v \in \mathcal{V}} \pi_v\, \mathcal{N}(s \mid \mu_v, \Sigma_v),$$

where $\mathcal{V}$ is the collection of Gaussian components and $\pi_v$ is the mixture weight. Instead of using source model softmax predictions $\theta(x)_z$ as in LEEP, they the GMM posterior distribution:

$$P(v \mid x) = P(v \mid s) = \frac{\pi_v\, \mathcal{N}(s \mid \mu_v, \Sigma_v)}{\sum_{v' \in \mathcal{V}} \pi_{v'}\, \mathcal{N}(s \mid \mu_{v'}, \Sigma_{v'})},$$

Substituting $\theta(x_i)_z$ in the LEEP formulation with $P(v \mid x_i)$, all remaining steps of the $\mathcal{N}$-LEEP computation proceed identically to LEEP.

**GBC (Pándy et al., 2022)** assumes per-class Gaussian distributions and assesses transferability by computing the Bhattacharyya coefficient (Bhattacharyya, 1946) between pairs of classes in the source model's embedding space. The probabilistic model for each class $p_c$ is defined as a Gaussian distribution $p_c = \mathcal{N}(\mu_c, \Sigma_c)$, with parameters estimated as follows:

$$N_c = \sum_i \mathbf{1}_{[y_i=c]},$$

$$\mu_c = \frac{1}{N_c} \sum_i \mathbf{1}_{[y_i=c]} f_s(x_i),$$

$$\Sigma_c = \frac{1}{N_c - 1} \sum_i \mathbf{1}_{[y_i=c]} (f_s(x_i) - \mu_c)(f_s(x_i) - \mu_c)^\top,$$

where $x_i$ denotes a target input, $f_s(x_i)$ denotes the feature representation extracted from the penultimate layer of the source pre-trained model, and $\mathbf{1}_{[y_i=c]}$ is the indicator function selecting samples belonging to class $c$. Thanks to the Gaussian assumption of the target class distributions, the Bhattacharyya distance $D_B$ between two classes $c_i$ and $c_j$ can be computed in closed form as:

$$D_B(c_i, c_j) = \frac{1}{8}(\mu_{c_i} - \mu_{c_j})^\top \Sigma^{-1}(\mu_{c_i} - \mu_{c_j}) + \frac{1}{2} \ln \left( \frac{|\Sigma|}{\sqrt{|\Sigma_{c_i}||\Sigma_{c_j}|}} \right),$$

where $\Sigma = \frac{1}{2}(\Sigma_{c_i} + \Sigma_{c_j})$. The Bhattacharyya coefficient between two class distributions is then computed as:

$$\mathrm{BC}(c_i, c_j) = \exp\left(-D_B(c_i, c_j)\right).$$

The final transferability score from source $s$ to target $t$ estimates the overall class overlap by summing the pairwise Bhattacharyya coefficients between all distinct target class pairs:

$$\mathrm{GBC}_{s \to t} = -\sum_{i \neq j} \mathrm{BC}(c_i, c_j),$$

where the negative sign ensures that higher GBC values correspond to lower overlap, indicating better transferability.

**Fréchet Inception Distance (FID) (Heusel et al., 2017; Fréchet, 1957)** measures the distance between two probability distributions, typically used to compare sets of images. Assuming the distributions follow multivariate Gaussian distributions, the distance takes a closed-form expression (Dowson & Landau, 1982). In practice, FID is computed on the encoded feature representations of the two datasets using a feature extractor (*e.g.*, a pre-trained Inception network). Let $\mu_X, \Sigma_X$ and $\mu_Y, \Sigma_Y$ be the means and covariance matrices of the encoded features from the two distributions $F$ and $G$, respectively. The Fréchet distance is given by:

$$d^2(F, G) = \|\mu_X - \mu_Y\|^2 + \mathrm{tr}\left[\Sigma_X + \Sigma_Y - 2\left(\Sigma_X \Sigma_Y\right)^{1/2}\right]$$

This formulation combines feature means and covariances providing a holistic measure of similarity between the two distributions. In transfer learning, it can serve to estimate the distance between source and target domains (Matsoukas et al., 2022) as a proxy to quantify domain similarity.

**Earth Mover's Distance (EMD)** Cui et al. (2018) estimate domain similarity between the source domain $S$ and target domain $T$ using Earth Mover's Distance (EMD) (Rachev, 1985; Rubner et al., 2000). To make computations tractable, they denote the source domain as $S = \{(s_i, w_{s_i})\}_{i=1}^m$ and the target domain as $T = \{(t_j, w_{t_j})\}_{j=1}^n$, where $s_i$ and $t_j$ represent the $i$-th and $j$-th categories in the source and target domains, respectively, and $w_{s_i}, w_{t_j}$ are the normalized number of images in each category such that

$\sum_{i=1}^{m} w_{s_i} = \sum_{j=1}^{n} w_{t_j} = 1$. They also define $g(s_i)$ and $g(t_j)$ as the mean of image features extracted from a generic feature extractor (*e.g.* the penultimate layer of a ResNet). The distance between the two domains is then computed using their Earth Mover's Distance (EMD):

$$d(S, T) = \frac{\sum_{i=1}^{m} \sum_{j=1}^{n} f_{i,j} \|g(s_i) - g(t_j)\|}{\sum_{i=1}^{m} \sum_{j=1}^{n} f_{i,j}},$$

where $f_{i,j}$ is the optimal flow from category $s_i$ to $t_j$ by minimizing the EMD optimization problem.

**Image Domain Similarity (IDS)**   Mensink et al. (2021) define a similarity metric between source $S$ and target $T$ datasets by extracting image features using a feature extractor. The domain difference is computed as the average distance from each target image to its closest source image:

$$D(T \mid S) = \frac{1}{|T|} \sum_{t} \min_{s} d(f_t, f_s),$$

where $|T|$ is the number of target images, $f_t$ and $f_s$ are the target and source image features, and $d(\cdot, \cdot)$ denotes the Euclidean distance. They compute this metric using 1000 randomly sampled images from each dataset.

