# OpenReview forum: "k-NN as a Simple and Effective Estimator of Transferability"
_TMLR — Accepted by TMLR_

### Review · Reviewer_yBPZ · 2025-06-10

**Summary Of Contributions:**

The setup of the paper is transfer learning, where informally one is interested in applying knowledge gained from data in one domain to improve the performance in another domain.
This work considers various metrics that are used to predict the transferability of a data domain.
Specifically, authors consider computer-vision datasets and compare the correlation between the transferability metrics and actual predictive performance (performance gain) in classification tasks.

The main finding of the paper is that the predictive performance of the $k$-NN classier in the target domain, whose similarity measure is formed by the features learned in the source domain, is positively correlated with the transfer performance.

**Audience:**

Yes

**Claims And Evidence:**

Yes

**Requested Changes:**

* There are a lot of textual citations where there should be parenthetic (e.g., all the metrics introduced in Section 2). These should be fixed.
* It would be more helpful to explain the math expressions (where possible) for the considered transferability metrics in appendix, since the reader needs to look at all these citations.

**Strengths And Weaknesses:**

## Strengths
The authors have conducted extensive experiments to investigate the predicability of transfer metrics in the literature.
The finding that the $k$-NN predictive performance serves as a transfer metric is interesting and could be practically useful given its simplicity. I consider this finding worth sharing.

## Weaknesses
Two points affecting the significance of the findings:
1. The paper only reports the correlation between the absolute/relative transferability performance and transferability metrics.
It would also be helpful to have the target values of absolute/relative transferability performance, since using these, we can identify in which problem settings, conducting knowledge transfer is useful.
While unlikely, it is conceivable that all the problem settings in the paper do not require any transfer (with low relative transfer performance), and the positive correlation of a transferability metric might not indicate that it is useful to predict when to transfer.

2. This point is related to the sample efficiency of transferability metrics, which is related to the practicality of transferability metrics. Transfer learning might be useful in a setting where the target domain does not have sufficiently large data to create a performant model (a situation where the relative transfer performance could be high).
It would be useful to know whether the $k$-NN method (or other metrics) could serve as a proxy of transferability in such low-target-data scenario --- the availability of the target data might affect transferability metrics differently.
While I realise that the transfer predictability is useful even outside this scenario (e.g., the target data is abundant, but the user might just want to have a performance boost), addressing this question would strengthen the paper.

While the paper is well written overall, there is room for improving the clarity
* It is ambiguous what kind of transfer learning is performed. My understanding here is that 'transfer learning' refers to using the neural network (up to the final layer) trained in the source domain as the initialisation for the network in the target domain. This point requires clarification.
* The words 'task' and 'domain' appear to be loosely defined. This is clarified somewhat in the paragraph in Section 4 (which metric... when only the domain changes), but in the first paragraph what these words mean. The authors might want to move the first paragraph (general predictors of transferability) to the end of the section.
* On a related note, I find the term "domain" confusing in the paragraph "Which metric is better when only the task changes?". Since in transfer learning, the domain always shifts (from the source to the target), the sentence 'domain remains the same' is a bit confusing. The point of this section is that the task complexity varies, and it seems redundant to say 'domain remains the same'.
* It seems all the experiments are based on computer-vision datasets. The paper's presentation should reflect on this point (as it can read like the findings are universal across different data types).


## Minor comments
* Typo: "it my not"
* Transfer gap is introduced and seems to have never been used in the paper.
* There are some words spelled in American English and others in British.

---

> ### Author Response · Authors · 2025-07-24
> **Response to Reviewer yBPZ**
>
> Thank you for your thoughtful and constructive review. Please find our responses to your comments below. We hope that our clarifications and the updates made to the manuscript adequately address your concerns. In light of these changes, we believe the paper now better aligns its claims with the supporting evidence, and we respectfully ask you to consider updating your "Claims and Evidence" rating to Yes.
>
> **1. Reporting target values of absolute/relative transfer performance**
> Below, we report the absolute and relative transfer learning performance as requested. We consider the general transfer learning setting where a ResNet-50, pre-trained on ImageNet, is transferred to various downstream datasets. We report absolute and relative transfer performance along with corresponding $k$-NN transferability estimates.
>
> |  | Dogs | CIF-100 | Aircraft | Birds | CIF-10 | APTOS | AID | Cal-101 | Cal-256 | Pets | DTD | SUN |
> | --- | --- | --- | --- | --- | --- | --- | --- | --- | --- | --- | --- | --- |
> | Abs. transfer perf | 0.83 | 0.83 | 0.79 | 0.69 | 0.96 | 0.91 | 0.89 | 0.93 | 0.83 | 0.91 | 0.68 | 0.61 |
> | Rel. transfer perf | 0.48 | 0.15 | 0.27 | 0.36 | 0.04 | 0.07 | 0.11 | 0.30 | 0.35 | 0.36 | 0.37 | 0.39 |
> | $k$-NN abs est | 0.81 | 0.59 | 0.20 | 0.32 | 0.82 | 0.68 | 0.74 | 0.86 | 0.75 | 0.90 | 0.61 | 0.49 |
> | $k$-NN rel est | 0.96 | 0.78 | 0.81 | 0.97 | 0.58 | 0.54 | 0.70 | 0.92 | 0.92 | 0.92 | 0.87 | 0.94 |
>
> As expected, some tasks show low relative transfer performance despite high absolute transfer performance (e.g., CIF-10 and APTOS). In cases where relative transfer performance is low, $k$-NN estimates also tend to be lower, although they may sometimes overestimate the actual transfer performance (transferability correlations are presented in Appendix Tables 4 and 5).
>
>
>
> **2. Sample efficiency**
> The requested analysis of $k$-NN in low-data scenarios is presented below. We evaluate the transferability of pre-trained architectures (details in Section 3) across various downstream tasks. Datasets are sorted based on their training set size. For each dataset, we report the performance of the $k$-NN metric (${\tau}_w$ correlation) in predicting absolute and relative transfer performance.
>
> |  | Cal-101 | APTOS | Pet | DTD | AID | Aircraft | Dogs | Cal-256 | SUN | Birds | CIF-10 | CIF-100 |
> | --- | --- | --- | --- | --- | --- | --- | --- | --- | --- | --- | --- | --- |
> | Train set Size | 3,030 | 3,113 | 3,680 | 3,760 | 5,000 | 6,667 | 12,000 | 15,420 | 23,929 | 19,850 | 50,000 | 50,000 |
> | $k$-NN corr with abs transfer perf | 0.43 | 0.47 | 0.34 | 0.79 | 0.31 | 0.36 | 0.88 | 0.82 | 0.93 | 0.45 | 0.57 | 0.84 |
> | $k$-NN corr with rel transfer perf | 0.04 | 0.08 | 0.65 | 0.39 | 0.58 | 0.46 | 0.70 | 0.87 | 0.61 | 0.50 | 0.52 | 0.43 |
>
> With the exception of predicting relative transfer performance on Cal-101 and APTOS, $k$-NN generally performs reliably, suggesting its effectiveness in low-data regimes, although the performance may vary depending on the dataset and task characteristics.
>
> **3. Points about clarity**
> - **Transfer learning type**: consistent with common practice in the literature, we fully fine-tune all the layers of the pre-trained network on the target domain. We have clarified this in Section 3 of the revised manuscript.
> - **Domain and task**: we have incorporated your suggestion and removed the phrase “domain remains the same” from the paragraph titled “Which metric is better when only the task changes?” Our intention in including the “General Predictors of Transferability” paragraph at the beginning was to address the most general transfer learning scenario—transferring to different downstream datasets. We then isolate the effects of task complexity and domain shift in the subsequent paragraphs to provide a more fine-grained analysis of metric performance across different scenarios.
> - **Paper’s presentation**: our work specifically focuses on image classification tasks. Please refer to our common response titled “Scope of our work.” We have updated the manuscript accordingly to clearly reflect this.
> - **Transfer gap**: The transfer gap is used to measure the correlation between domain distance metrics and the gains from transfer learning. Since there is an inherently inverse relationship between domain distance and transfer learning gain—where a larger domain distance typically corresponds to a smaller transfer learning gain—we define the transfer gap to explicitly capture this inverse relationship. We have clarified this in the updated manuscript.
> - **Typos and spellings**: Thank you for pointing them out. They have been corrected in the revised manuscript.
>
> **4. Requested changes**
> - **Textual citations**: We have now corrected the citation style in the revised manuscript.
> - **Math expressions**: As requested, we have included the mathematical formulations of representative metrics with concise formulations in Appendix I of the updated manuscript.

---

### Review · Reviewer_jtYo · 2025-07-08

**Summary Of Contributions:**

This paper systematically benchmarks 23 transferability metrics and introduces k-NN as a simple yet superior alternative. The authors demonstrate k-NN’s effectiveness across diverse settings and provide a clear understanding of transferability in deep learning.

**Audience:**

Yes

**Broader Impact Concerns:**

/

**Claims And Evidence:**

Yes

**Requested Changes:**

1. Although the empirical results are strong, the theoretical explanation for why k-NN outperforms other metrics is inadequate. Despite being discussed within a PAC learning framework, the work still lacks a solid theoretical foundation.
2. All experiments are confined to image classification. The study does not cover broader transfer learning tasks such as object detection, segmentation, or text domains, which may limit the generality of the conclusions.

**Strengths And Weaknesses:**

1. The paper presents a large-scale empirical validation of over 42,000 experiments, covering 16 datasets and 23 mainstream transferability metrics. The large scale and breadth of the evaluation are compelling.
2. The authors provide a well-structured taxonomy of transferability metrics, categorizing them into domain distance metrics, metrics under architectural changes, and metrics between source and target tasks.
3. The authors propose to employ k-NN as a transferability metric and demonstrate its outstanding performance in predicting both absolute and relative transfer outcomes.  The proposed algorithm shows strong robustness to hyperparameter choices.
4. The paper distinguishes and investigates three different transfer factors—domain, task, and architecture—and provides a thorough analysis of how existing metrics perform under different factors.
5. The authors simultaneously consider both absolute and relative transfer performance and especially highlight the importance of comparing transfer learning against random initialization. This perspective significantly enhances the validity of the conclusions.

---

> ### Author Response · Authors · 2025-07-24
> **Response to Reviewer jtYo**
>
> Thank you for providing valuable feedback on our submission. Please find our responses to your comments below:
>
> **1. Theoretical Foundation**
> Please refer to our common response “Theoretical Analysis.” We would like to emphasize that our work is primarily empirical rather than theoretical, with a focus on comprehensive experimentation to provide practical and reliable insights. Theoretical analysis of full fine-tuning is often infeasible or limited in scope, which makes empirical evaluation essential for understanding transferability in real-world settings..
>
> **2. Broader Tasks**
> Please refer to our common response titled “Scope of Our Work.” Additionally, we have revised the manuscript to clarify the scope of our study.

---

### Review · Reviewer_MQRB · 2025-07-16

**Summary Of Contributions:**

This paper explores the most effective transfer learning metric. The authors argue that the good transfer learning metric should not only consider the accuracy in predicting the final performance of the transferred model in the new setting, but also the ability to predict performance improvements gained through the transfer learning process. Based on this demand, the authors conduct an extensive evaluation involving over 42,000 experiments comparing 23 transferability metrics across 16 different datasets to assess their ability to predict transfer performance and reveal that none of the existing metrics perform well across the board. Furthermore, k-nearest neighbor evaluation is found to be the best metric.

**Audience:**

Yes

**Broader Impact Concerns:**

None.

**Claims And Evidence:**

Yes

**Requested Changes:**

See Weaknesses.

**Strengths And Weaknesses:**

Strengths:
1. Extensive experiments show the effectiveness of KNN on image classification.
2. The definition of the ideal transfer learning metric is meaningful.

Weaknesses:
1. Although extensive studies show the effectiveness of KNN, this paper still lacks the empirical analysis on explanation (e.g., examples' visualization in the low dimension), and the theoretical analysis is also not complete on many metrics.
2. This paper cannot show how KNN further pushes the transfer learning community. For example, how to guide the application in practical scenarios.
3. The finding is interesting, but the contribution to methods is limited.
4. Is this conclusion effective on other tasks, such as text classification?

---

> ### Author Response · Authors · 2025-07-24
> **Response to Reviewer MQRB**
>
> Thank you for your insightful and constructive review. Below, we provide detailed responses to your comments and questions.
>
> **1. Analysis on explanation and theory**
> We thank the reviewer for the suggestion. As requested, we have added visualizations of feature representations across multiple downstream tasks in Appendix H of the updated manuscript. These t-SNE plots offer qualitative insights into how feature space structure correlates with transfer performance. While visualizations help build intuition, our focus is on systematic quantitative evaluation, which we believe offers a more objective and broadly applicable understanding of metric performance.
> Regarding the theoretical analysis, we kindly refer the reviewer to our common response titled “Theoretical analysis”.
>
> **2. Pushing the transfer learning community**
> For each transfer factor, we provided detailed and actionable guidelines on which metrics perform best under the specific conditions. We propose $k$-NN as a new, robust, and practically viable transferability metric, introduced with real-world applicability as a core goal. We believe these contributions offer clear value to the transfer learning community by enhancing both the reliability and usability of transferability estimation in practical settings. The experimental scenarios we designed—including variations in task, domain, and architecture—were intentionally chosen to reflect realistic transfer learning use cases. Our work draws meaningful and novel practical insights from this large-scale, systematic experimentation.
>
> **3. Method contribution**
> Our paper makes a methodological contribution by proposing $k$-NN as a new transferability metric and extensively validating its effectiveness. We believe this is a valuable contribution to the community.
>
> **4. Text classification**
> Please refer to our common response titled “Scope of Our Work.”

---

### Author Response · Authors · 2025-07-24
**General Comment**

We sincerely thank the reviewers for their thoughtful reviews and constructive feedback.
We appreciate **Reviewer MQRB**'s recognition of our extensive experiments and finding our definition of an ideal transfer learning metric meaningful. We are grateful to **Reviewer jtYo** for finding the scale and breadth of our study compelling, noting our thorough analysis of transferability factors, recognizing our taxonomy of metrics as well-structured, and acknowledging the robustness of our proposed $k$-NN approach. They noted that our perspective of considering both absolute transfer performance and relative transfer performance significantly enhances the validity of our conclusions. We thank **Reviewer yBPZ** for recognizing the extensiveness of our experiments and for acknowledging our findings interesting and worth sharing.


**Scope of our work**
Reviewers inquired whether our findings extend to other tasks such as segmentation, detection, or natural language processing (NLP). Our study primarily focuses on image classification, which is the central task in computer vision for evaluating transferability metrics. To the best of our knowledge, the application of existing transferability metrics to tasks such as segmentation or detection is not yet well established.
We acknowledge that parts of the original manuscript may have implied broader applicability, and we have clarified the scope accordingly in the revised version.


**Theoretical analysis**
We note that our work is empirical and experimental in nature, with the primary goal of defining the desiderata for a robust transferability metric, rather than conducting a theoretical investigation. Theoretical analysis is often difficult or only applicable in simplified cases, making empirical evaluation essential for practical insights into transferability. We made an effort to provide a PAC learning interpretation to offer additional insight—particularly to compare $k$-NN with its most similar metrics—highlighting its advantages through its non-parametric nature (Appendix G). Ultimately, we believe that experimental evaluation provides the most reliable basis for assessing transferability and providing practical guidelines. To this end, we have conducted an extensive analysis of transfer learning across a wide range of experimental configurations.

---

### Decision · Action_Editor_Jab2 · 2025-09-01

**Recommendation:** Accept as is

**Audience:**

Yes

**Audience Explanation:**

A large part of the community is actively experimenting with image classification, and transfer learning remains a highly important paradigm. The AE agrees with the reviewers that this paper provides valuable insights on the topic and makes a meaningful contribution to the community.

**Claims And Evidence:**

Yes

**Claims Explanation:**

This paper presents an extensive study on transferability metrics for image classification. The authors conduct a large number of experiments and evaluate a wide range of existing metrics from the literature. They also show that a simple k-NN-based metric performs as well as, or better than, most established alternatives. The authors made changes following reviewer suggestions and clarified the scope better.

All three reviewers agree that the experiments are thorough and that the empirical evidence sufficiently supports the paper’s claims. The main weaknesses raised are:
a) the scope is limited to image classification, and
b) there is no theoretical analysis.
However, neither of these points was considered sufficient to justify rejection. While the study is indeed focused solely on image classification, this is common practice in the field. Moreover, the paper is positioned as a primarily empirical contribution, and the strength of the experimental results supports its value.